

# Temporal and spatial evaluation of satellite-based rainfall estimates across the complex topographical and climatic gradients of Chile

Mauricio Zambrano-Bigiarini[1,2], Alexandra Nauditt[3], Christian Birkel[4,5], Koen Verbist[6,7], and Lars Ribbe[3]

[1]Department of Civil Engineering, Universidad de La Frontera, Temuco, Chile
[2]Center for Climate and Resilience Research, Universidad de Chile, Santiago, Chile
[3]Institute for Technology and Resources Management in the Tropics (ITT), Cologne Technical University, Betzdorfer Strasse 2, 50679 Köln, Germany
[4]Department of Geography, University of Costa Rica, 2060 San José, Costa Rica
[5]Northern Rivers Institute, University of Aberdeen, AB24 3UF Aberdeen, UK
[6]Hydrological Systems & Global Change Section, UNESCO Chile, Santiago, Chile
[7]International Centre for Eremology, Department of Soil Management, Ghent University, Ghent 9000, Belgium

*Correspondence to:* Mauricio Zambrano-Bigiarini (mauricio.zambrano@ufrontera.cl)

**Abstract.** Accurate representation of the real spatio-temporal variability of catchment rainfall inputs is currently severely limited. Moreover, spatially interpolated catchment precipitation is subject to large uncertainties, particularly in developing countries and regions which are difficult to access (e.g., high elevation zones). Recently, satellite-based rainfall estimates (SRE) provide an unprecedented opportunity for a wide range of hydrological applications, from water resources modelling to

monitoring of extreme events such as droughts and floods.

This study attempts to exhaustively evaluate -for the first time- the suitability of seven state-of-the-art SRE products (TMPA 3B42v7, CHIRPSv2, CMORPH, PERSIANN-CDR, PERSIAN-CCS-adj, MSWEPv1.1 and PGFv3) over the complex topography and diverse climatic gradients of Chile. Different temporal scales (daily, monthly, seasonal, annual) are used in a point-to-pixel comparison between precipitation time series measured at 366 stations (from sea level to 4600 m a.s.l. in the Andean

Plateau) and the corresponding grid cell of each SRE. The modified Kling-Gupta efficiency was used to identify possible sources of systematic errors in each SRE. In addition, several categorical indices were used to assess the ability of each SRE to correctly identify different precipitation intensities.

Results revealed that most SRE products performed better for the humid South (36.4-43.7°S) and Central Chile (32.18-36.4°S), in particular at low- and mid-elevation zones (0-1000 m a.s.l.) compared to the arid northern regions and the Far

South. Seasonally, all products performed best during the wet seasons (MAM-JJA) compared to summer (DJF) and autumn (SON). In addition, all SREs were able to correctly identify the occurrence of no rain events, but they presented a low skill in classifying precipitation intensities during rainy days. Overall, PGFv3 exhibited the best performance everywhere and for all time scales, which can be clearly attributed to its bias-correction procedure using 217 stations from Chile. Good results were also obtained by CHIRPSv2, TMPA 3B42v7 and MSWEPv1.1, while CMORPH, PERSIANN-CDR and PERSIANN-CCS-adj

were not able to represent observed rainfall. While PGFv3 (currently available up to 2010) might be used in Chile for historical analyses and calibration of hydrological models, the high spatial resolution, low latency and long data records of CHIRPS and



TMPA 3B42v7 (in transition to IMERG) show promising potential to be used in meteorological studies and water resources assessments. We finally conclude that despite improvements of most SRE products, a site-specific assessment is still needed before any use in catchment-scale hydrological studies.

## 1 Introduction

Accurate representation of the real spatio-temporal variability of catchment rainfall inputs is currently severely limited. Traditionally, precipitation (P) data are collected through ground-based observations using rain gauges and/or weather radars. Catchment-representative rainfall is usually obtained by interpolation of point rainfall measured at rain gauges (e.g. Rogelis and Werner, 2013; Verworn and Haberlandt, 2011; Zhang and Srinivasan, 2009; Kurtzman et al., 2009). However, even in densely monitored regions, precipitation data are highly uncertain (Tian and Peters-Lidard, 2010; Woldemeskel et al., 2013). In developing countries and regions which are difficult to access -such as high altitudes- there is usually only a sparse network of meteorological stations available, and therefore the obtained spatial rainfall fields are subject to even larger uncertainties (Woldemeskel et al., 2013). Furthermore, raingauge rainfall measurements are affected by wind, flawed installation, wetting losses, and other random and systematic errors (e.g. Ren and Li, 2007; Sevruk and Chvíla, 2005).

To overcome some of the aforementioned limitations of ground-based rainfall measurements, space-based estimates of precipitation provide a promising alternative source. Several near-global high resolution satellite-based rainfall estimates (*SRE*) have recently become operational, including the Precipitation Estimation from Remotely Sensed Information using Artificial Neural Networks (PERSIANN; Sorooshian et al. 2000; Hsu et al. 1997), the PERSIANN-Cloud Classification System estimation (PERSIANN-CCS; Hong et al. 2004), the National Oceanic and Atmospheric Administration (NOAA) Climate Prediction Center morphing technique product (CMORPH; Joyce et al. 2004; Janowiak et al. 2005), and the Tropical Rainfall Measuring Mission (TRMM) Multi-satellite Precipitation Analysis products (TMPA, Huffman et al. 2007), among others. The instruments on-board the satellites include passive microwave (PMW), visible (VIS) and infra-red (IR) sensors, and meteorological radar. Recent advances in sensor technology and methods for merging various data sources (e.g., geostationary thermal infra-red, passive microwave, radar, and information from the Global Telecommunication System (GTS)) have led to a continuous improvement of temporal and spatial resolution of these precipitation products (e.g. Kidd et al., 2009). Satellite estimates of precipitation have received different names and acronyms in the literature: satellite precipitation estimates (*SPE*, Scofield and Kuligowski 2007), satellite-based rainfall estimates (*SRFE*, Thiemig et al. 2012), satellite quantitative precipitation estimates (*SQPE*, Lee et al. 2015) and satellite rainfall estimates (*SRE*, Abera et al. 2016). Here, we will use *SRE* to name satellite-based rainfall estimates throughout the text.

The emergence of the aforementioned near-global and high-resolution SRE opens up new possibilities for applications in data-scarce or ungauged regions. However, SRE products need evaluation and often calibration before any use in hydrological applications. Tian and Peters-Lidard (2010) studied uncertainties in SRE, by computing the variance from an ensemble of six different TRMM data sets. They found that SRE are more reliable over areas with strong convective precipitation and flat surfaces, such as the tropical oceans and South America. Dinku et al. (2010) evaluated CMORPH and two TMPA products (3B42



and 3B42RT) for mountainous regions of Africa and South America. Both TMPA products underestimated the occurrence and amount of rainfall which they attributed to the complex terrain and orographic rain process. Scheel et al. (2011) compared TMPA 3B42-V6 estimates with rain gauges in the regions of Cuzco (Peru) and La Paz (Bolivia). They detected large biases in the estimation of daily precipitation amounts. The occurrence of strong precipitation events was well represented but their

intensities were underestimated. In addition, TMPA estimates for La Paz showed high false alarm ratios. Thiemig et al. (2012) compared six SRE against rain gauge data over four African river basins (Zambezi, Volta, Juba-Shabelle, and Baro-Akobo). They found that SRE showed higher performance over the tropical wet and dry zone compared to semiarid mountainous regions, low accuracy in detecting heavy rainfall events over semiarid areas, general underestimation of heavy rainfall events, and overestimation of the number of rainy days in the tropics. Demaria et al. (2011) used an object-based verification method

to explore the existence of systematic errors for three SRE in South America (La Plata River Basin): TRMM, CMORPH and PERSIANN. They found that PERSIANN underestimated the observed average rainfall rate and maximum rainfall, CMORPH overestimated the average rainfall rate while the maximum rainfall was slightly underestimated, and the average rainfall rate and volume provided by TRMM correlated well with ground observations, whereas the maximum rainfall was systematically overestimated. In general, there does not seem to exist a single SRE product that performs best always and everywhere. There-

fore, the performance of each SRE needs to be assessed for each individual case study. Furthermore, there is little guidance on which performance criteria to use in evaluating SRE products.

Nevertheless, climate and hydrological studies in data-sparse regions can benefit from the spatial coverage and grid-structure of SRE to drive hydrological simulations for water resources management (e.g. Tobin and Bennett, 2014; Meng et al., 2014; Xue et al., 2013; Li et al., 2012; Khan et al., 2011; Hong et al., 2009; Su et al., 2008; Thiemig et al., 2013; Xue et al., 2013),

mapping of natural hazards (e.g. Werren et al., 2016; Hong et al., 2007), and drought risk assessment (e.g. Tao et al., 2016; AghaKouchak et al., 2015; Zhang and Jia, 2013; Naumann et al., 2012). Reliable information on the spatio-temporal variability of rainfall is also one of the main factors to achieve food security, in particular in data scarce regions (Kang et al., 2009; Verdin et al., 2005).

Notwithstanding all the recent improvements of SRE, there are still some issues that need to be addressed before reaching

its full potential (Tobin and Bennett, 2014). Several researchers have found that satellite products fail in capturing certain types of precipitation events, have important biases and present false detection of precipitation (e.g. Gebregiorgis and Hossain, 2013; Thiemig et al., 2012; Ebert et al., 2007). When SRE products are used to drive hydrological models, the aforementioned errors are propagated in a non-linear way to simulated streamflows (see e.g. Bisselink et al., 2016; Nikolopoulos et al., 2010; Fekete et al., 2004). In the worst case, such simulations may lead to wrong conclusions and poor management decisions with po-

tentially devastating societal consequences. In addition, different hydrological applications, such as drought monitoring, flood forecasting, water resources management or allocation of long-term water rights, require precipitation at very different time scales, ranging from hourly to seasonal (Tobin and Bennett, 2014). Also, there is a clear need to better understand how hydrological simulations forced by satellite-derived data depends on the climatological regime and time scale used for simulations (see e.g. Gebregiorgis and Hossain, 2013; Nikolopoulos et al., 2010; Tobin and Bennett, 2014).





In the last decade, droughts of unusual severity have affected the Chilean territory, both because of their intensity and multi-annual duration (Boisier et al., 2016). This is consistent with climate change projections for that region, which indicate central-southern Chile as a global hotspot for increased drought frequency, with likely water security issues in this region (Prudhomme et al., 2014). On the other hand, flood events have a relatively normal occurrence in Chile (Müller et al., 2011) and some particularly intense events have affected the country in recent years. Considering the significant socio-economic costs of extreme water-related hazards (Blume et al., 2008; Favier et al., 2009) we chose the mountainous Chilean territory as case study, due to the presence of extreme elevations (0-6893 m a.s.l.) and its heterogeneous hydroclimatic conditions (hyperarid in the north and extremely wet in the south). Further, there is almost no climate monitoring being carried out above 1500 m a.s.l.. The present study attempts to exhaustively evaluate -for the fist time- the suitability of seven state-of-the-art SRE products for hydrological application in this data-scarce and complex mountainous region. Spatio-temporal characteristics of different SRE products are compared against that of existing rain gauge data, by using different continuous and categorical performance measures. In particular, this study will address the following research questions:

1. what is the overall performance of each SRE?

2. which SRE performs best across the topographic and climate gradient in Chile?

3. which SRE performs best for different time scales (daily, monthly, seasonal, annual)?

4. how does the accuracy of a given SRE change for different precipitation intensities?,

5. when a SRE does not capture the observed precipitation, is it due to a misrepresentation of the shape, magnitude, variability or all the previous properties of the precipitation time series?, and

6. is there any SRE that performs best compared to all the others, everywhere and for all time scales?

Results of this study aim at increasing our knowledge about the suitability of different SRE estimates to characterise the spatio-temporal distribution of precipitation across the climatically and topographically diverse Chilean territory, as a cost-effective complement to ground-based measurement networks in this data-scarce region. In addition, findings of this study aim at providing feedback to the developers of different SRE products for potential use in future releases.

The article is organized as follows: Section 2 presents the study area and datasets, with Section 3 describing the methodology used to compare satellite products against ground observations. Numerical and graphical results are shown in Section 4, whereas Section 5 provides an in-depth discussion in the light of the wider literature. We present concluding remarks in Section 6.





## 2 Study area and datasets

### 2.1 Study area

The continental area of Chile has more than 4000 km of latitudinal extension, from 17.50°S to 66.42°S, bounded by the Pacific Ocean to the west ( 76° W) and by the Andes mountain range in the east ( 66° W). Four main morphological units

condition the existence of eleven different types of climate (from hot dessert to polar/tundra) and associated vegetation: Coastal Plains, Coastal Mountains, Intermediate Depression and the Andes, with elevations ranging from sea level to 6893 m a.s.l. at the Ojos del Salado volcano. Figure 1 shows the location of the study area, including a digital elevation model and main river basins (panel a), mean annual precipitation (panel b) and temperature (panel c), and the eleven climate types identified based on Köppen classification. The main factors affecting the climate of the Chilean territory are the latitude, topography

and the oceanic influence coming from the long Pacific Ocean (INE, 2015). The northern part of Chile is characterized as hyperarid/arid/semiarid, with extremely low precipitation and high temperatures; while abundant precipitation is observed in the south, reaching amounts of up to 5000 mm/year with lower temperatures (Valdés-Pineda et al., 2014). The four traditional seasons of the southern hemisphere are present in Chile: autumn (MAM), winter (JJA), spring (SON), and summer (DJF), with the wet season occurring predominantly during winter in most of the central-southern territory, during summer in the regions of

Tarapacá and Antofagasta (*Bolivian winter*), and no clearly defined dry season in the southernmost part of Chile (INE, 2015). An exhaustive description and references about the type of precipitations present in Chile and the global patterns that influence its interannual variability can be found on Valdés-Pineda et al. (2014).

    To evaluate the performance of SRE, five major macroclimatic zones in Chile were selected, as shown in Figure 1: (i) Far North (17.50-26.00°S), (ii) Near North (26.00-32.18°S), (iii) Central Chile (32.18-36.40°S), (iv) South (36.40-43.70°S) and

(v) Austral/Far South (43.70-56.00°S), slightly adapted from DGA (2016).

### 2.2 Datasets

### 2.2.1 Rain gauges

Time series of observed precipitation were downloaded from an updated dataset of 781 rain gauges provided by the Center of Climate and Resilience Research (CR2), with daily data from 01-Jan-1940 to 31-Dec-2015 (available on http://www.cr2.

cl/recursos-y-publicaciones/bases-de-datos/). The original raw data were provided by the Chilean Water and Meteorological agencies (DGA and DMC, respectively). However, in order to use only stations with a long record of observations but also to ensure a minium number of stations representative of the Chilean topography and climatic zones, we selected only rain gauges with less than 2% of missing data during the period Jan/2003-Dec/2010, which resulted in 366 stations within our study area (see Section 3.1). Figure 1 a shows the spatial distribution of the selected rain gauges. Daily time series in all the 781 stations

analysed in this work can be found in the supplementary material (Zambrano-Bigiarini et al., 2016).

    It is worth mentioning that several SREs use observed precipitation data from the Global Precipitation Climatology Centre (GPCC) to adjust their satellite-only estimates. Here we analysed the number of precipitation gauges from Chile used in the



GPCC dataset. Figure 2 shows that this number has fluctuated over time, reaching a maximum of 146 in the period 1969-1995, dropping to a value of around 30 stations in the last two decades. The spatial distribution of the gauges used to create the GPCC for Chile has also observed a strong contraction (Figure 2), leaving a large area of the territory without any observation, which subsequently increases the estimation errors of the final product.

### 2.2.2 Satellite-based data

Seven of the most state-of-the-art SREs with at least 10 years of daily estimates and relatively high spatial resolution are compared against observed precipitation. A brief description of each SRE with some previous applications is given in the next paragraphs, but the interested reader can find more information in the references provided for each product.

**CMORPH**

The NOAA Climate Prediction Center (CPC) MORPHing technique (Joyce et al., 2004) provides quasi-global estimates of precipitation at relatively high spatial resolution ($0.07° \times 0.07°$, which is about 8 km at the equator) and frequent temporal resolution (half-hourly and 3-hourly), from 60°N to 60°S. CMORPH estimates are based solely on PMW data (Joyce et al., 2004; Janowiak et al., 2005) with IR imagery not used to estimate precipitation but only to interpolate between two PMW-derived rainfall intensity fields. CMORPH has been reported to outperform other SRE products over the Australian tropics (Ebert et al.,

2007; Joyce et al., 2004), central United States (Behrangi et al., 2011), and Europe. As many SRE products that do not rely on rainfall gauge data, CMORPH tends to overestimate the amount of precipitation during wet periods (Behrangi et al., 2011). Pereira Filho et al. (2010) compared CMORPH rainfall estimates over South America (Amazon Basin), at 8-km spatial resolution, with available rainfall observations at daily, monthly, and yearly accumulation time scales. Their results show that the correlation between satellite-derived and gauge-measured precipitation increases with the accumulation period, from daily to

monthly, especially during the rainy season.

**PERSIANN-CDR**

The PERSIANN algorithm uses an Artificial Neural Network (ANN) model to estimate precipitation using IR. Its accuracy is improved by adaptive adjustment of the network parameters using rainfall estimates from a passive microwave sensor. At the pixel level, the algorithm fits the mean and standard deviation of the brightness temperature of a pixel and the adjacent pixel's

temperature texture to the calculated precipitation rate (Hsu et al., 1997). A new product named The PERSIANN-CDR (for Climate Data Record) was developed by applying the PERSIANN algorithm to Gridded Satellite Infrared Data (GridSat-B1) and then bias-correcting estimations using 2.5° monthly GPCP precipitation data (Ashouri et al., 2015). This product provides a 30-year data of near-global (60°S-60°N) daily precipitation data at 0.25° spatial resolution. Mei et al. (2014) compared measured precipitation data in mountainous regions of the Italian Alps with PERSIANN-CDR and found that the product slightly

overestimated P in regions with low rainfall and underestimated P in regions with high rainfall. All the daily data for the study area were downloaded from http://www.climatedatalibrary.cl/SOURCES/.UCIrvine/.CHRS/.PERSIANN-CDR.



**PERSIANN-CCS-adj**

PERSIANN-CCS (Hong et al., 2004) is an infrared-based satellite estimation process, employing image processing and pattern recognition techniques to develop a patch-based cloud classification system (CCS). Combined with an artificial neural network model, hourly pixel precipitation intensity is estimated globally at 0.04°x0.04° and accumulated daily. The PERSIANN-CCS-adj was developed especially for Chile applying a non-parametric Quantile Mapping and Gaussian Weighting (QM-GW) interpolation process to reduce the systematic biases existing in the daily PERSIANN-CCS dataset over the country. In summary, for the adjusted PERSIANN-CCS the cumulative density functions of 456 observed rain gauges for different seasons were used to effectively correct the biases in the PERSIANN-CCS rainfall estimates. This increases the consistency of spatial rainfall patterns and removes systematic biases at monthly and daily timescales. A full description of the methodology and validation results can be found in Yang et al. (2016).

**TMPA-3B42v7**

The Tropical Rainfall Measuring Mission (TRMM) Multi-satellite Precipitation Analysis (TMPA) is intended to provide a "best" estimate of quasi-global precipitation from a wide variety of modern satellite-borne precipitation-related sensors. Rainfall estimates are provided at relatively high spatial resolution (0.25°x0.25°) at 3-hourly time steps, in both real and post-real time, to meet a wide range of research needs (Huffman et al., 2010). This SRE combines IR data from geo-stationary satellites of four passive microwave (PMW) sensors, namely TRMM Microwave Imager (TMI), Special Sensor Microwave/Imager (SSM/I), Advanced Microwave Sounding Unit-B (AMSU-B), and the Advanced Microwave Scanning Radiometer-EOS (AMSR-E). The TMPA products include the version 6 (V6) and version 7 (V7) real-time products 3B42RT (3B42RT-V6 and 3B42RT-V7), the 3-hourly research products 3B42 (3B42-V6 and 3B42-V7), and the monthly products 3B43 (3B43-V6 and 3B42-V7). The 3B42 algorithm is executed in four steps: 1) PMW precipitation estimates are calibrated and combined; 2) IR precipitation estimates are generated using the calibrated PMW data; 3) both IR and PMW data are then combined; and 4) rescaled on a monthly basis using two sources of raingauge data: the GPCP monthly rain gauge analysis developed by the Global Precipitation Climatological Center and the Climate Assessment and Monitoring System (CAMS) monthly rain gauge analysis developed by CPC (Huffman et al., 2010). The newest version of TMPA 3B42 (V7) was released on June 2012, and recent studies show that 3B42-V7 estimates improve upon 3B42-V6 (e.g. Chen et al., 2013). Today, after more than 17 years of data collection, the instruments on TRMM were turned off on April 8th 2015, but the TMPA 3B42 product will continue to be produced through early 2018 (https://pmm.nasa.gov/data-access/downloads/trmm).

**CHIRPS**

The Climate Hazards Group InfraRed Precipitation with Station data (CHIRPS v.2) is a global daily, pentadal and monthly precipitation product explicitly designed for monitoring agricultural drought and global environmental change over land (Funk et al., 2015). CHIRPS combines remotely sensed precipitation of geo-synchronous and polar-orbiting satellites, from five different products (such as the Climate Prediction Center MORPHing Technique (CMORPH) and sensors such as the Tropical Rainfall Measurement Mission (TRMM), with more than 2000 station records to calibrate global Cold Cloud Duration (CCD) rainfall estimates (Funk et al., 2015). The product features a spatial resolution of 0.05° from 50°S to 50°N (across all longi-



tudes) with a > 30 years final monthly precipitation record (1981-2015). The station data from the near real-time World Meteorological Organization's Global Telecommunication System (GTS) are continuously used to update roughly every two days (a stable product is released every three weeks) and validate the remote sensors using additional information (physiographic and remotely sensed earth energy emissions that correspond to the location and intensity of precipitation) via mowing-window

geostatistical regression (Funk et al., 2015). In summary, the CHIRPS process involves three main steps: i) the Climate Hazards group Precipitation climatology (CHPclim Funk et al. 2015), ii) the satellite-only Climate Hazards group Infrared Precipitation (CHIRP), and iii) the station calibration procedure. Even though, the continuous development of CHIRPS is mainly in support of drought-related issues in Africa (Climate Hazards Group), there are now other global applications available (e.g., http://ewx.chg.ucsb.edu:8080/EWX/index.html, http://chg.geog.ucsb.edu/tools/geowrsi/index.html) and also papers that have

looked at climate dynamics in South America (Ceccherini et al., 2015; Deblauwe et al., 2016).

**MSWEP**

In June 2016 a new global precipitation dataset was released, the Multi-Source Weighted-Ensemble Precipitation (MSWEP) dataset providing data for the period 1979-2015 with a 3-hourly temporal and 0.25° spatial resolution (Beck et al., 2016). It was specifically developed for hydrological modelling with the aim to overcome shortages related to the performance of satellite

products in representing precipitation in mountainous, tropical and snowmelt driven regions. It is based on different types of data sources as rain-gauge measurements, satellite observations as well as estimates from atmospheric models. The long-term mean of the MSWEP dataset was based on the CHPclim dataset. Where available, data were replaced by more accurate regional datasets. A correction for gauge under-catch and orographic effects was introduced by inferring catchment-average P from streamflow (Q) observations at 13 762 stations across the globe. The temporal variability of MSWEP was determined by

weighted averaging of P anomalies from seven datasets; two based solely on interpolation of gauge observations (CPC Unified and GPCC), three on satellite remote sensing (CMORPH, GSMaP-MVK, and TMPA 3B42RT), and two on atmospheric model reanalysis (ERA-Interim and JRA-55). For each grid cell, the weight assigned to the gauge-based estimates was calculated from the gauge network density, while the weights assigned to the satellite- and reanalysis-based estimates were calculated from their comparative performance at the surrounding gauges. The MSWEP dataset was validated at the global scale using

the conceptual HBV light rainfall-runoff model with different precipitation data from 9011 catchments (<50000 km$^2$) across the globe. MSWEP obtained the highest daily correlation coefficient (R) among the five P datasets for 60.0% of the stations and a median R of 0.67 versus 0.44-0.59 for the other datasets and a median calibrated NSE of 0.52 versus 0.29-0.39 for other P datasets.

**PGF**

The PGFv3 product is an improvement over the Princeton University global meteorological forcing (PGF) dataset (Sheffield et al., 2006). In brief, the PGF dataset merges data from the National Centers for Environmental Prediction-National Center for Atmospheric Research (NCEP-NCAR) reanalysis (Kalnay et al., 1996) with the Global Precipitation Climatology Project (GPCP; Adler et al. 2003), Tropical Rainfall Measuring Mission (TRMM) Multisatellite Precipitation Analysis (TMPA; Huffman et al. 2007), observation-based datasets of precipitation, and the Climatic Research Unit (CRU) precipitation dataset





(Harris et al., 2013). To ensure adequate representation of rain day anomalies, a correction is applied by resampling daily precipitation data to match the statistics of observation-based precipitation datasets (CRU, GPCP). Downscaling from 2.0° to 1.0° uses a Bayesian probability function considering the fraction of wetted area using a higher resolution dataset (GPCP). The daily precipitation values are also bias-corrected by matching the NCEP-NCAR monthly totals with the monthly values of the

CRU dataset. Additional improvements are described in Chaney et al. (2014), performing a spatial downscaling from 1.0° to 0.25° resolution using bilinear interpolation involving 128 stations over Chile from the Global Surface Summary of the Day Version 7 dataset archived at the National Climate Data Center (GSOD, ftp://ftp.ncdc.noaa.gov/pub/data/gsod). The version 3 of the PGF used here was finally obtained by merging 217 local precipitation station datasets (obtained from the Chilean Water Agency - DGA) into the PGF using a Kalman filter approach (Chaney et al., 2014; Peng et al., 2016), for the period 1979-2010.

As such, the spatial coverage was greatly improved, as well as the statistical precipitation characteristics (frequency, amount and extreme values) at different temporal scales. The PGF is integrated into the Princeton Latin American Flood and Drought Monitor - LAFDM (http://stream.princeton.edu/) and can be directly downloaded.

All the aforementioned SREs ingest some type of observed precipitation data in the algorithm used to compute the rainfall estimates, except CMORPH. Among the SREs that use observed precipitation data, we identified two groups: those products

that use mostly GPCC observed data to produce the satellite estimates (PERSIANN-CDR, 3B42v7, CHIRPSv2, MSWEPv1.1) and those SREs that use a specific set of Chilean rain gauges to bias-correct their satellite estimates (PGFv3 and PERSIANN-CCS-Adj).

Despite that some of the aforementioned products are available at sub-daily resolution (e.g., CMORPH and TMPA 3B42v7), in this study we follow Abera et al. (2016) and only used the daily products for two main reasons: (i) there are no time

series of observed precipitation available at sub-daily time scales with enough data length, (ii) pointing at future hydrological simulations for allocation of water resources and drought monitoring tasks, the daily time scale is suitable for capturing the temporal variation of streamflows at basin scales. Table 1 provides a summary of the satellite-based datasets, including their full names, and spatial and temporal resolutions for the versions used in this study.

## 3    Assessment of precipitation products

### 25   3.1    Methodology for data comparison

Following Thiemig et al. (2012), a point-to-pixel analysis was applied to compare time series of data observed at selected rain gauges (red circles in Figure 1) to the corresponding SRE pixel. A comparison at sub-basin spatial scales was not carried out due to the lack of rain gauges in the upper Andes, which would involve large uncertainties in any interpolation based on existing point measurements. All SRE products with a spatial resolution lower than 0.25° were upscaled to a unified grid of 0.25°, in

order to enable a consistent point-to-pixel comparisons. Daily precipitation events were classified and analysed adapting the criteria defined by the World Meteorological Organization (2008), as shown in Table 2, because the same amount of rainfall





falling with two different durations will lead to different hydrological processes in a given catchment. Daily observations at the 366 rain gauges (see Section 2.2.1) and corresponding satellite estimates were accumulated into monthly, seasonal (DJF, MAM, JJA, SON) and annual values, to assess the accuracy of each precipitation product at different time scales. Based on the availability of satellite data (see Table 1), the evaluation period for this study extends from Jan 2003 to Dec 2010.

## 3.2 Performance indices

An exhaustive evaluation of the seven SREs described in Section 2.2.2 was carried out, using both continuous and categorical indices of model performance at different temporal scales. Continuous indices are described in Appendix A, and they include the modified Kling-Gupta efficiency (Kling et al., 2012; Gupta et al., 2009) along with its three individual components. The modified Kling-Gupta efficiency (*KGE'*, dimensionless) is a relatively new index to compare observations with estimations, which is used here to decompose the total performance of the SREs into a linear correlation ($r$, Equation A2), bias ($\beta$, Equation A3) and a variability ($\gamma$, Equation A4) term. We selected the $KGE'$ because forecasting and hydrological applications generally require that rainfall estimates are able to reproduce the temporal dynamics (measured by $r$) as well as preserving the volume and distribution of precipitation (measured by $\beta$ and $\gamma$, respectively). The optimum value of *KGE'*, $r$, $\beta$ and $\gamma$ is 1.0. The Pearson product-moment correlation coefficient $r$ is a measure of the linear correlation between the observations and satellite values, ranging from +1 (perfect positive correlation) to -1 (perfect negative correlation), with zero values indicating absence of linear correlation. The main drawback of $r$ is its inability to detect changes in location and scale between the two variables. $\beta$ measures the average tendency of the satellite values to be larger ($\beta > 1$, overestimation) or smaller ($\beta < 1$, underestimation) than their observed counterparts. $\gamma$ indicates if the dispersion of the satellite estimates is higher or lower compared to the observations. Using $CV_s/CV_o$ for the computation of the variability ratio, instead of using only the $\sigma_s/\sigma_o$, we ensure that the bias and variability ratios are not cross-correlated (Kling et al., 2012).

Most of the studies assessing SREs against ground observations separate rainfall events into *no rain* and *rain* to evaluate the skill of each SRE in reproducing such events (e.g. Guo et al., 2015; Blacutt et al., 2015; Ward et al., 2011; Scheel et al., 2011). Here, we further subdivide daily rainfall in five types of events, which are used to classify precipitation based on its daily intensity, ranging from *no rain* (*dry day*) ($< 1$ mm d$^{-1}$) to *violent rain* ($> 40$ mm d$^{-1}$), as shown in Table 2. Five categorical indices, described in Appendix B, were used to assess the ability of each satellite product to correctly identify the five aforementioned categories/classes of daily rainfall events (see Table 2). The *percent correct* (*PC*, Equation B1) is a simple measure that indicates the percent of events and no-events that are correctly identified, ranging from zero (absence of (no-)events correctly identified) to one (all (no-)events correctly identified). The *PC* is not useful for low frequency (extreme) events, because misleading high values of the score are usually obtained due to the high frequency of correct negative ($CN$) events. This shortcoming is compensated for by the next three scores. The *probability of detection* (*POD*, Equation B2, also known as *hit rate*) and the *false alarm ratio* (*FAR*, Equation B3) measure the fraction of events that are correctly and incorrectly identified by the satellite product, respectively. Both indices, *POD* and *FAR*, range from 0 to 1, but 1 is the optimum value for *POD* while a *FAR=0* indicates that no events are incorrectly identified by the SRE. The *equitable threat score* (*ETS*, Equation B4), also known as *Gilbert's Skill Score* (*GSS*), measures the fraction of observed and/or estimated events that were correctly





predicted, adjusted by the frequency of hits that would be expected to occur simply by random chance (for example, it is easier to correctly match rain occurrence in a wet climate than in a dry climate). ETS ranges from -1/3 to 1, being 1 its optimal value and scores below 0 indicate that the chance forecast of the event should be preferred over the actual unskilled SRE value. The *frequency bias* (*fBias*, Equation B5) compares the number of events identified by the satellite product to the number of events

that actually occurred at the corresponding raingauge. It is commonly referred to as *bias* when there is no possible confusion with other meanings of the term (not in this article). The optimal value of *fBias* is 1.0 (unbiased), i.e., the event was registered by the SRE the same number of times than at the raingauge, with $fBias > 1$ indicating an overestimation of the occurrences by the SRE, whereas $fBias < 1$ reveals that the event was identified by the SRE less times than it was actually observed at the raingauge.

Due to the large amount of numerical and graphical results produced in the current analysis, only categorical values corresponding to $POD$ and $fBias$ will be presented and discussed in Section 4, as they provide a good summary of the detection capabilities of each SRE product. A well-performing satellite product should have a value of *FAR* close to zero, and values of *KGE'*, $r$, $\beta$, $\gamma$, *PC*, *POD*, *ETS*, and *fBias* close to 1. All the aforementioned indices of SRE performance were computed based on Jolliffe and Stephenson (2003), using the R environment 3.3.1 (R Core Team, 2015) and the `raster` (Hijmans, 2016),

`hydroGOF` (Zambrano-Bigiarini, 2016a) and `hydroTSM` (Zambrano-Bigiarini, 2016b) R packages.

## 4    Results

### 4.1    Spatial variability of SRE performance

We computed spatial maps showing the values of continuous indices of SRE performance for each of the 366 selected stations and for different time scales (daily, monthly, annual, four seasons), resulting in 196 different figures of the Chilean territory

(7 SREs x 7 time scales x 4 indices of model performance). We put special emphasis on showing SRE performance at different elevation zones and latitudes. Figure 4 is one example of the aforementioned maps, showing the modified Kling-Gupta efficiency ($KGE'$) with CHIRPS estimates compared against observed precipitation at a monthly time scale. It showed a good performance ($KGE' > 0.75$) in low and mid elevation areas (0-1000 m a.s.l.) of Central and Southern climate zones (32.18-43.70°S) and acceptable performance in the Near North area (26.0-32.18°S). However, it showed a poor performance

($KGE' < 0$) for the high lands of the Andean Plateau (*Altiplano*), specifically between 2000-3500 m a.s.l. To disentangle the origin of differences between observed precipitation and CHIRPS estimates, Figures 5 to 7 showed the spatial variation of the individual components of monthly $KGE'$. First, Figure 5 showed a good linear correlation ($r \geq 0.75$) in most of the Chilean territory, with a few exceptions in the Far North area (17.5-26.0°S). Secondly, Figure 6 revealed that in most of the territory the satellite estimates present a low bias ($0.75 \leq \beta \leq 1.25$), with exception of stations located in the Far North, presenting a large

overestimation. In particular, over the Andean Plateau satellite estimates were as high as 94 times the observed precipitation (note that average annual values for those stations is, in general, lower than 1 mm). Finally, Figure 7 showed that CHRIPS slightly underestimated the variability of observed monthly precipitation ($0.75 \leq \gamma \leq 1$) of almost all the stations, with larger values only at the high elevations zones (2000-4600 m a.s.l.) of the Far North. Figures similar to 4-7 can be found on the sup-





plementary material Daily time series in all the 781 stations analysed in this work can be found in the supplementary material (Zambrano-Bigiarini et al., 2016) for the other six SREs and time scales.

As a way of summarizing the findings for different macro-climatic areas, Figure 8 shows boxplots with the modified Kling Gupta efficiency between different monthly SREs and their corresponding observations. It clearly illustrates that the best per-

formance for all SREs but CMORPH was obtained in the South (36.4-43.7°S), followed by the Central (32.18-36.4°S) and Austral areas (43.7-56°S), with PGF as the best product followed closely by CHIRPS, MSWEP and 3B42v7, while CMORPH presented the lowest median performance in those climatic areas. On the other hand, all SREs except PERSIANN-CDR presented an acceptable performance in the Near North (26.0-32.18°S) and a poor performance in the Far North (17.5-26.0°S). Similarly to the previous figure, but focusing on elevation zones instead of macro-climatic areas, Figure 9 shows that the best

performance for all SREs was obtained in low- and mid-elevation zones (0-1000 m a.s.l.). On the other hand, all SREs except PGF performed poorly for the higher elevations, in particular between 2000 and 3500 m a.s.l. The best SRE product was PGF, followed closely by CHIRPS, 3B42v7 and MSWEP, while CMORPH presented the lowest median performance in those elevation areas (Figure 9).

## 4.2 Temporal variability of SRE performance

Figures 10 to 13 show boxplots summarizing the modified Kling-Gupta efficiency ($KGE'$) and its individual components of performance ($r, \beta, \gamma$) for each satellite product and for different time scales.

Figure 10 shows the Pearson product-moment correlation coefficient ($r$) between different satellite estimations and the observations at the corresponding grid cell, for seven different temporal scales: daily, monthly, annual, summer (DJF), autumn (MAM), winter (JJA), and spring (SON). It depicts that correlations are positive, with most daily values ranging from 0.2 to

0.6, and achieving higher median values at monthly (0.60-0.95) than at annual (0.3-0.9) time scales. In general, all the products presented the highest correlation during autumn (MAM) and the lowest values during summer (DJF) and spring (SON). The PGF product was an outlier, which exhibited median correlation values larger than 0.9 for all time scales, followed by MSWEP, CHIRPS, 3B42v7 and PERSIANN-CDR.

Figure 11 depicts the bias ratio of the modified Kling-Gupta efficiency ($\beta$) for different satellite estimations against obser-

vations at the corresponding grid cell, and for the same temporal scales used before. PGF, CHIRPS and 3B42v7 were almost unbiased for all time scales and seasons except for the summer (DJF), for which almost all the products overestimated P compared to the observed precipitation. MSWEP and PERSIANN-CDR tended to overestimate P compared to the observed precipitation for all time scales, while CMORPH and PERSIANN-CCS-Adj showed a general underestimation thereof.

Figure 12 shows the variability ratio of the modified Kling-Gupta efficiency ($\beta$) between SREs and the observations. Most of

the products resulted in underestimating the variability of P for all time scales, with exception of CMORPH and PERSIANN-CCS-Adj which overestimated the observed variability at annual and winter (JJA) time scales. 3B42v7 and PGF are the products that best resembled the variability of observed precipitation, while PERSIANN-CDR is the one with the largest median underestimation at all time scales.





Figure 13 summarizes the three previous figures into one, showing boxplots of $KGE'$ for all the satellite products and time scales. Most of the SREs, except PGF, presented a limited overall performance at daily time scales (median values of $KGE' \leq 0.3$), which improved when aggregated into monthly values (median values of $KGE' \geq 0.5$). PGF and CMORPH were the products with the best and worst performance for almost all scales, while 3B42v7, CHIRPS and MSWEP showed the

second best performance. At the seasonal scale all the products performed best during autumn (MAM) and winter (JJA), and showed the worst performance during spring (SON) and summer (DJF), with PERSIANN-CDR and MSWEP being the worst.

### 4.3 SRE performance for different precipitation intensities

Figures 15 and 16 show boxplots summarising the categorical performance indices used to identify the capabilities of each SRE to capture the five types of precipitation events described in Table 2. Figure 15 presents boxplots with the probability of

detection $POD$, to provide an in-depth look at the high values of percent of correct ($PC$) obtained by all the SREs in all the precipitation intensities (not shown here). Most of the products showed a low $POD$, in general with values lower than 0.6, for all the rainfall events except by *no rain* ($[0, 1)$ mm d$^{-1}$), which was well captured by all SREs ($POD >= 0.8$). PGF is the product that best captures all classes of rainfall events, while CMORPH resulted in the worst performance for events larger than 5 mm day$^{-1}$, with almost no skill in detecting rainfall events larger than 20 mm day$^{-1}$. Figure 16 shows that the amount of

*no rain* events identified by most of SREs was in good agreement with the corresponding observed amount, with MSWEP and PERSIANN-CDR showing a small underestimation. All SREs overestimated the amount of light rain events, $[1, 5)$ mm day$^{-1}$, and underestimated the number of violent rain events, $> 40$ mm day$^{-1}$. In between, MSWEP, CHIRPS, 3B42v7, PERSIANN-CCS-Adj and PERSIANN-CDR over-estimated moderate rainfall events,$[5, 20)$ mm day$^{-1}$, while CMORPH underestimated its amount and PGF provided an unbiased estimation of the number of this type of events; heavy rainfall events,$[20, 40)$ mm

day$^{-1}$, were well captured by all the SREs but CMORPH and PERSIANN-CDR, which largely underestimated its amount. All the boxplots presented in this section and those corresponding to the other categorical indices described in Appendix B (not shown here) can be found in the supplementary material Daily time series in all the 781 stations analysed in this work can be found in the supplementary material (Zambrano-Bigiarini et al., 2016).

### 5 Discussion

This work aimed at providing, for the first time, insights about the performance of seven state-of-the-art SREs (TMPA 3B42v7, CHIRPSv2, CMORPH, PERSIANN-CDR, PERSIAN-CCS-adj, MSWEPv1.1 and PGFv3) at different temporal scales (daily, monthly, seasonal, annual) over the Chilean territory, using 366 stations located from sea level to 4600 m a.s.l. in the Andean Plateau.




### 5.1 Overall performance of the evaluated SREs

As expected, the two products that used a Chilean dataset for calibration showed a good performance at all time scales and nation-wide, with PGFv3 showing an outstanding performance in comparison to all SREs, while PERSIANN-CCS-adj had a performance more similar to the other SREs (Figure 13).

Overall, PGFv3 was the best performing product in terms of $KGE'$ and its individual components (see Figures 10 to 13), which was expected due to the use of 217 local raingagues in the bias correction procedure used to adjust its estimates. PGFv3 was followed closely by CHIRPSv2, TMPA 3B42v7 and MSWEPv1.1, all of them using the GPCC dataset to calibrate their precipitation estimates, not any local dataset. Generally, SREs performed better for wetter periods (i.e., MAM-JJA, Figure 13) and southern and central regions (Figure 8). All SREs except PGF performed poorly for the higher elevations, in particular between 2000 and 3500 m a.s.l. (Figure 9) and also for the Far North desert region (Figure 8).

### 5.2 Which SRE performs best across the topographic and climate gradient in Chile?

Mountainous regions pose important challenges across all seasons to satellite estimates derived from both TIR and PM observations (Tian and Peters-Lidard, 2010; Dinku et al., 2010). Figure 9 illustrated that all SREs except PGFv3 performed poorly for the higher elevations, in particular between 2000 and 3500 m a.s.l. However, for the few stations located in the highest elevations (3500-4600 m a s.l.), results are better, especially for PERSIANN-CCS-Adj and PGFv3 which can be explained by the use of Chilean datasets in the bias-correction procedure of these products. On the other hand, an extreme variation was observed for the results along the latitudinal gradient, covering eleven types of climate, from hot dessert to polar/tundra (Figure 1). Figure 8 showed that all SRE perform best in the humid South (36.4-43.7°S), followed by Central Chile (32.18-36.4°S) and Far South (43.7-56.0°S), compared to the arid northern regions, in particular in the desert Far North where satellite estimates were as high as 94 times the observed precipitation (e.g., Figure 6 for CHIRPSv2). Time series comparison between rain gauges and SRE data can be found on the supplementary material Daily time series in all the 781 stations analysed in this work can be found in the supplementary material (Zambrano-Bigiarini et al., 2016) for each one of the 366 selected stations described in Section 2.2.1.

### 5.3 Which SRE performs best for different time scales?

The PGF exhibited the best $KGE'$ performance for all time scales, followed by CHIRPS, 3B42v7 and MSWEP, while the worst performing were CMORPH, PERSIANN-CCS-Adj and PERSIANN-CDR (Figure 13). In line with previous work by Scheel et al. (2011) and Pereira Filho et al. (2010), better performance was obtained for monthly time scales compared to the daily scale. Unexpected, however, was the fact that the annual scale resulted worse compared to the daily scale, in particular for CMORPH and PERSIANN-CCS-Adj, which very likely is due to the effect of monthly to annual temporal aggregation, where small differences between monthly SRE values and observations during a large number of months are then summed up when aggregated into a smaller number of annual values. Figure 13 illustrated that most SREs showed a distinct behaviour during different seasons, performing better during the more humid seasons (MAM-JJA) than during the dry ones (SON-DJF). How-





ever, while PGF, CHIRPS, 3B42v7 and CMORPH presented minor differences in performance along the seasons, MSWEP, PERSIANN-CDR and PERSIANN-CCS-Adj showed large seasonal differences and large spread of performance during summer (DJF) and spring (SON).

### 5.4 How does the accuracy of a given SRE change for different precipitation intensities?

Evaluating the SRE performance for different precipitation intensities resulted in low values of most categorical performance indices. This was partly due to the demanding classification criteria: we used 5 types of precipitation classes instead of the typical two classes (rain/no-rain) (e.g. Dinku et al., 2010; Scheel et al., 2011). All SREs were able to correctly identify the occurrence of no rain events, but during rainy days they presented a low skill in providing an accurate classification of precipitation intensities (Figure 15). On the other hand, all SREs underestimated the number of violent rain events, $> 40$ mm day$^{-1}$ and overestimated the amount of light rain events, $[1, 5)$ mm day$^{-1}$ (Figure 16). In between, PGF provided an unbiased estimation of the number of moderate rainfall events,$[5, 20)$ mm day$^{-1}$, while CMORPH underestimated that amount and MSWEP, CHIRPS, 3B42v7, PERSIANN-CCS-Adj and PERSIANN-CDR overestimated it. The number of heavy rainfall events,$[20, 40)$ mm day$^{-1}$ was well captured by all the SREs but CMORPH and PERSIANN-CDR, which largely underestimated its amount.

### 5.5 How well does the aggregated KGE and its components evaluate SRE performance?

Modified Kling-Gupta efficiency ($KGE'$, Kling et al. 2012; Gupta et al. 2009) proved to be a useful index to assess the performance of SRE products, because it captures in a single number a pseudo multi-objective evaluation of the linear correlation, bias and variability of the satellite estimations compared to its corresponding observations at rain gauges.

Figure 10 showed that linear correlation ($r$) between SRE values and their corresponding observations were in general acceptable, with higher values at monthly times scales (0.60-0.95) than at annual (0.3-0.9) and daily (0.35-0.9) ones. In general, all the SREs presented the highest correlation during autumn (MAM) and the lowest values during summer (DJF) and spring (SON). Figure 11 showed that PGF, CHIRPS and 3B42v7 were almost unbiased ($\beta$) for all time scales and seasons except for the summer (DJF), where almost all the products overestimated P. CMORPH and PERSIANN-CCS-Adj showed a general underestimation of P, while MSWEP and PERSIANN-CDR tended to overestimate it. The aforementioned underestimation of CMORPH was in agreement with previous studies (e.g. Abera et al., 2016; Ringard et al., 2015; Lo Conti et al., 2014; Dinku et al., 2010) as well as the acceptable linear correlation (Abera et al., 2016), in particular at monthly time scales. The fact that CMORPH presented the highest underestimation of rainfall for almost all time scales very likely is because CMORPH does not use any observed precipitation data to compute its estimates, in contrast to all the other analysed SREs. Figure 12 illustrated that most of the products underestimated the variability of P for all time scales, with exception of CMORPH and PERSIANN-CCS-Adj which overestimated the observed variability at annual and winter (JJA) time scales. PGF and 3B42v7 were the products that best captured the variability of observed precipitation, while PERSIANN-CDR presented the largest median underestimation at all time scales. From the analysis described above, it is clear that providing a better representation of the variability of observed precipitation should be a major concern in future releases of the analysed SREs. In particular, PGF, CHIRPS and 3B42v7 which presented high linear correlation and were almost unbiased for monthly and annual scales.



## 5.6 Is there any SRE that performs best compared to all the others, everywhere and for all time scales?

PGv3F vas the best product for all time scales, macro-climatic areas and elevation zones, which was expected because this product used 217 rain gauges to bias-correct the original PGF estimates using a Kalman filter approach. Therefore, it is not entirely fair to compare PGF to other products that did not use local data for calibration. However, PGF was included as
potentially useful for some hydrological application in the Chilean territory. Interesting is the case of PERSIANN-CCS-Adj, which also used observed rain gauges, but which was unable to obtain results similar to those of PGF. The latter might be due to the use of a non-parametric Quantile-Mapping and Gaussian Weighting (QM-GW) algorithm to bias-correct the original PERSIANN-CCS estimates (Yang et al., 2016)). However, PERSIANN-CCS-adj is still under development through improved QM-GW calibration procedures using monthly cumulative distribution functions, as well as a real-time assimilation of precip-
itation measurements to improve performance at daily and monthly timescales in the near future.

## 6 Conclusions

Satellite-based rainfall estimates (SRE) provide an unprecedented opportunity to be applied in a wide range of meteorological and hydrological applications. Despite that most of the existing products are continuously improving their algorithms and data sources to adapt to particular environments, catchment-specific studies should still be carried out before any hydrological
application thereof.

In this article, the performance of seven state-of-the-art SRE datasets (3B42v7, CHIRPSv2, CMORPH, PERSIANN-CDR, PERSIAN-CCS-adj, MSWEPv1.1, PGFv3) was compared against observations from 366 rain gauges over the Chilean territory, which is extremely challenging due to its complex topography (elevations ranges from sea level to 6893 m a.s.l.) and the existence of 11 different types of climates (from hot dessert to polar/tundra) along the large latitudinal extent of 4300 km. Seven
different temporal scales relevant for hydrological applications were addressed in the evaluation: daily, monthly, annual, DJF (summer), MAM (autumn), JJA (winter), SON (spring). The modified Kling-Gupta efficiency ($KGE'$) was used as continuous index of performance, along with its three individual components (linear correlation, bias and variability), to identify possible sources of systematic errors in each SRE. In addition, several categorical indices ($PC$, $POD$, $FAR$, $fBias$, $ETS$) were used to assess the ability of each SRE to correctly identify five different intensities of daily precipitation.
Results revealed that the best performing SRE product was the PGFv3, calibrated with a Chilean data set, followed closely by CHIRPSv2, TMPA 3B42v7 and MSWEPv1.1, while CMORPH presented the lowest median performance everywhere, which was expected because it is the only product that does not use any type of measured rainfall for calibration. All SREs except PGFv3 performed poorly for the higher elevations, in particular between 2000 and 3500 m a.s.l. and also for the extreme Northern desert region. Surprisingly, all products performed worse for the annual scale compared to the monthly scale and even
daily scale.

The following paragraphs summarise the key findings:





- Most SRE products performed better for the humid South (36.4-43.7°S) and Central Chile (32.18-36.4°S), in particular at low- and mid-elevation zones (0-1000 m a.s.l.), compared to the arid northern regions and the Far South.

- Most SREs performed worst in the high elevation areas ($\geq 2000$ m a.s.l.) of the hyper-arid Far North (17.5-26.0°S).

- All SRE present positive values of linear correlation with ground observations, with best median values at monthly time scales ($0.4 \leq r \leq 0.95$) than at daily and annual scales. Correlations are also higher during autumn (MAM) and lower during summer (DJF) and spring (SON).

- PGFv3, CHIRPSv2 and TMPA 3B42v7 are almost unbiased at all time scales (except summer). MSWEPv1.1 and PERSIANN-CDR tend to overestimate P compared to observations, while CMORPH and PERSIANN-CCS-Adj tend to underestimation at all time scales.

- All SREs underestimated the observed variability of P at all time scales, except for CMORPH and PERSIANN-CCS-Adj which overestimated P at the annual scale and winter (JJA). TMPA 3B42v7 and PGF are the products that best captured the observed variability of P, while PERSIANN-CDR presented the poorest performance.

- $KGE'$ and its three individual components are recommended for further comparisons, because they do not only provide an overall assessment of the SRE performance, but also allowed us to understand if mismatches are due to errors in the representation of shape, magnitude or variability of the observed precipitation. If detected errors are mostly due to under or over-estimation of observed precipitation, it is very likely that applying a bias-correction procedure will significantly improve the performance of the product (e.g., PGFv3). Providing a better representation of the variability of observed precipitation should be a major concern in future releases of the analysed SREs, in particular for PGF, CHIRPS and 3B42v7 which presented high linear correlations with observed P and were almost unbiased at monthly and annual time scales.

- All SREs performed best in terms of $KGE'$ during the wet seasons autumn and winter (MAM-JJA) compared to summer (DJF) and autumn (SON).

- Overall, the best SRE product was PGFv3 followed by CHIRPSv2, TMPA 3B42v7 and MSWEPv1.1, which is expected because PGF used 217 Chilean rain gauges in its bias-correction procedure used to obtain the final estimations.

- All SREs were able to correctly identify the occurrence of no rain events (i.e., dry days). However, during rainy days they presented a low skill in providing an accurate classification of precipitation intensities.

- All SREs overestimated the occurrence of light rain events ([1, 5] mm d$^{-1}$), while underestimated the amount of violent rain events ($> 40$ mm d$^{-1}$). High precipitation events were well captured by all products.

- In general, CMORPH, PERSIANN-CDR and PERSIANN-CCS-Adj were considered not suitable to be used in hydrological applications in Chile at this point in time, due to their large biases when compared to observed precipitation records.




– Lack of rain gauges at higher elevation zones (over 2000 m a.s.l.) over most of the Chilean territory (south of 26.0°S), prevented from an exhaustive assessment of SRE products in such areas. The same applied for mid-elevation zones (200-1000 m a.s.l.) in the Far North (17.0-26.0°S) and Far South (southern to 43.7°S).

**Outlook**

Despite continuous improvements of most SRE products, a site-specific calibration is still recommended before any use in hydrological studies. This was evidenced by the PGFv3 dataset that showed a better performance compared to the other products due to the statistical merging of local precipitation observations. This highlights the need for adequate station observation networks to complement the SRE products, due to the dependency of SRE product quality on this ground-truthing. This is especially valid for high elevation areas, where only limited precipitation records are available, and where well-calibrated SRE

products could provide the highest added-value.

To conclude, the PGFv3 is the best performing product for Chile at all time scales and locations tested, but it is currently only available up to 2010, making it a relevant dataset for water balance estimations but not for real-time applications. Nevertheless, it is expected that PGFv3 will be used to bias-correct the TMPA 3B42v7 as part of the Latin American Flood and Drought Monitor. Since the PGFv3 dataset is performing well for all the indices tested, it is expected that the calibrated TMPA 3B42v7

will also improve its performance. As this process is still under development, it could not be tested in this study and will become the subject of future research. For (quasi)real-time and hydrological applications, we suggest to consider the use of CHIRPSv2 , because it has a long data record (1981-present), a low latency (1 day-3 weeks, depending on the product), and high spatial resolution (0.05° instead of the 0.25° of most SREs). MSWEPv1.1 is also a promising product, but it is only available up to 2014 with a long latency period. The TMPA 3B42v7 has a lower spatial resolution and covers a shorter period

(1998-present), but showed a very similar performance to CHIRPSv2 across all indices. The 3B42 data set will be superseded by the Global Precipitation Measurement (GPM) mission product IMERG, launched by NASA and the Japanese Aerospace Exploration Agency (JAXA) on February 27th 2014. To overcome the short data length of the new IMERG dataset, it is planned to provide a retrospectively processed IMERG datasets from 1998 onward, which are expected to be released in early 2017 (Huffman, 2015).

Given the high dependency of most SRE products on the GPCC dataset for calibration, the authors would like to recommend to the Chilean authorities in charge of the collection and analysis of meteorological data (e.g., DGA and DMC) to make additional efforts to share their data with GPCC, in order to improve the performance of SREs that ingest observed precipitation data.

**Appendix A: Continuous indices of model performance**

$$KGE = 1 - \sqrt{(r-1)^2 + (\beta-1)^2 + (\gamma-1)^2} \tag{A1}$$





$$r = \frac{\sum_{i=1}^{n}(O_i - \bar{O})(S_i - \bar{S})}{\sqrt{\sum_{i=1}^{n}(O_i - \bar{O})^2}\sqrt{\sum_{i=1}^{n}(S_i - \bar{S})^2}} \tag{A2}$$

$$\beta = \frac{\mu_s}{\mu_o} \tag{A3}$$

$$\gamma = \frac{CV_s}{CV_o} = \frac{\sigma_s/\mu_s}{\sigma_o/\mu_o} \tag{A4}$$

where $N$ is the number of observations; $O_i$ and $S_i$ are the observed and the corresponding satellite precipitation values at day
$i$, respectively; $\overline{O}$ and $\overline{S}$ are the arithmetic mean of the observations ans satellite estimates, respectively; $O_{max}$ and $O_{min}$ are
the maximum and minimum observed value, respectively.

**Appendix B: Categorical indices of model performance**

$$PC = \frac{H + CN}{Ne} \tag{B1}$$

$$POD = \frac{H}{H + M} \tag{B2}$$

$$FAR = \frac{FA}{H + FA} \tag{B3}$$

$$ETS = \frac{H - H_e}{(H + F + M) - H_e} \tag{B4}$$

$$fBias = \frac{H + F}{H + M} \tag{B5}$$

where $Ne$ is the total number of verification points (number of events), $H$ indicates a *hit*, i.e., a satellite estimate that correctly
identifies the type of rainfall event measured at the rain gauge; $M$ is a *miss*, i.e., an event recorded at the rain gauge but
not correctly identified by the satellite product; $F$ is a *false alarm*, i.e., a rainfall event detected by the satellite product but
not recorded at the corresponding rain gauge; $CN$ represents the *correct negatives* (or *correct rejections*),i.e., an event not
registered either by the rain gauge or the satellite; and $H_e$ indicate a *hit that could occur by chance*, computed as $H_e = (H + M)(H + F)/Ne$.





*Acknowledgements.* The authors thank to FONDECYT 11150861 "Understanding the relationship between the spatio-temporal characteristics of meteorological drought and the availability of water resources, by using satellite-based rainfall and snow-cover data. A case study in a data-scarce Andean Chilean catchment" for financial support to the main author; the FONDAP 15110009 for providing the precipitation dataset used in this study; to the Center for Hydrometeorology and Remote Sensing at the University of Irvine for providing the PERSIANN-CCS-adj and PERSIAN-CDR datasets, the Terrestrial Hydrology Research Group at Princeton University for providing the PGFv3 datasets; to Thomas Cram (NCAR/CISL/DSS) for technical support about CMORPH data, Daniel Wunder (NOAA) for technical support related to PERSIANN-CDR, Hylke Beck for providing an updated version 1.1 of MSWEP data. The main author is also grateful to the active R community for unselfish and prompt support, in particular to Robert J. Hijmans for developing and maintaining the `raster` R package. In the framework of this special issue, the authors also wish to thank Eric Wood, Justin Sheffield and Liqing Peng from the Terrestrial Hydrology Research Group at Princeton University for the local calibration of the PGF dataset for Chile and its integration into the Latin American Flood and Drought Monitor.



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





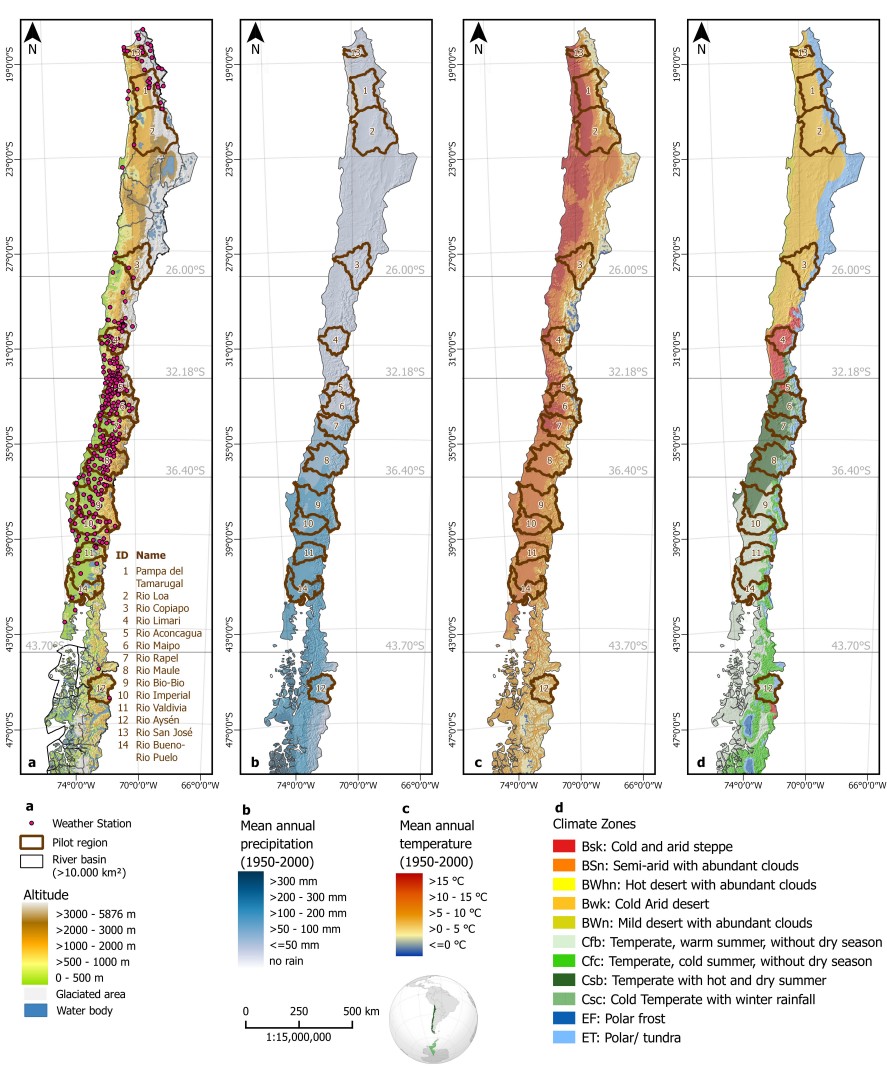

**Figure 1.** Study Area. From left to right: (a) digital elevation model (Jarvis et al., 2008), main Chilean basins, and location of 289 selected rain gauge; (b) mean annual precipitation (Hijmans et al., 2005); (c) mean annual temperature (Hijmans et al., 2005); and (d) climate zones based on Köppen classification.





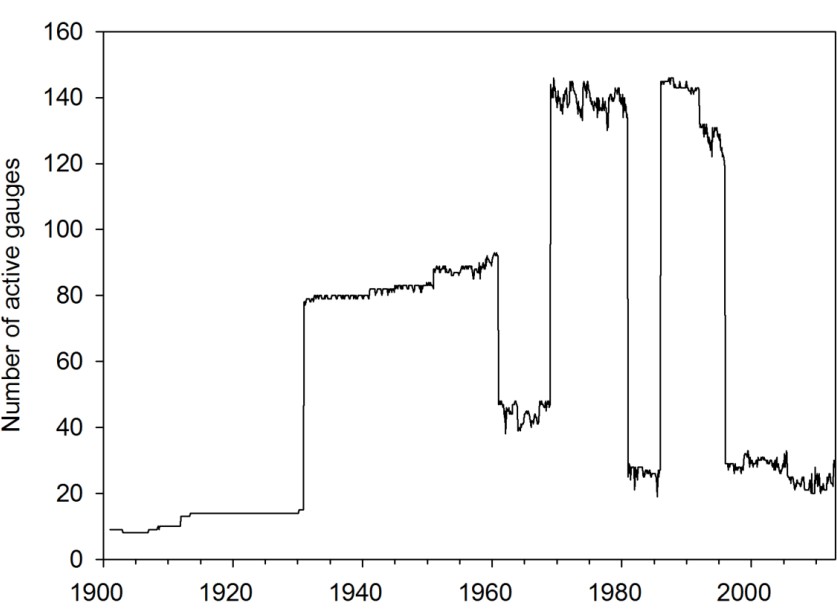

**Figure 2.** Number of rain gauges in Chile used in the GPCC dataset (Schamm et al., 2015).





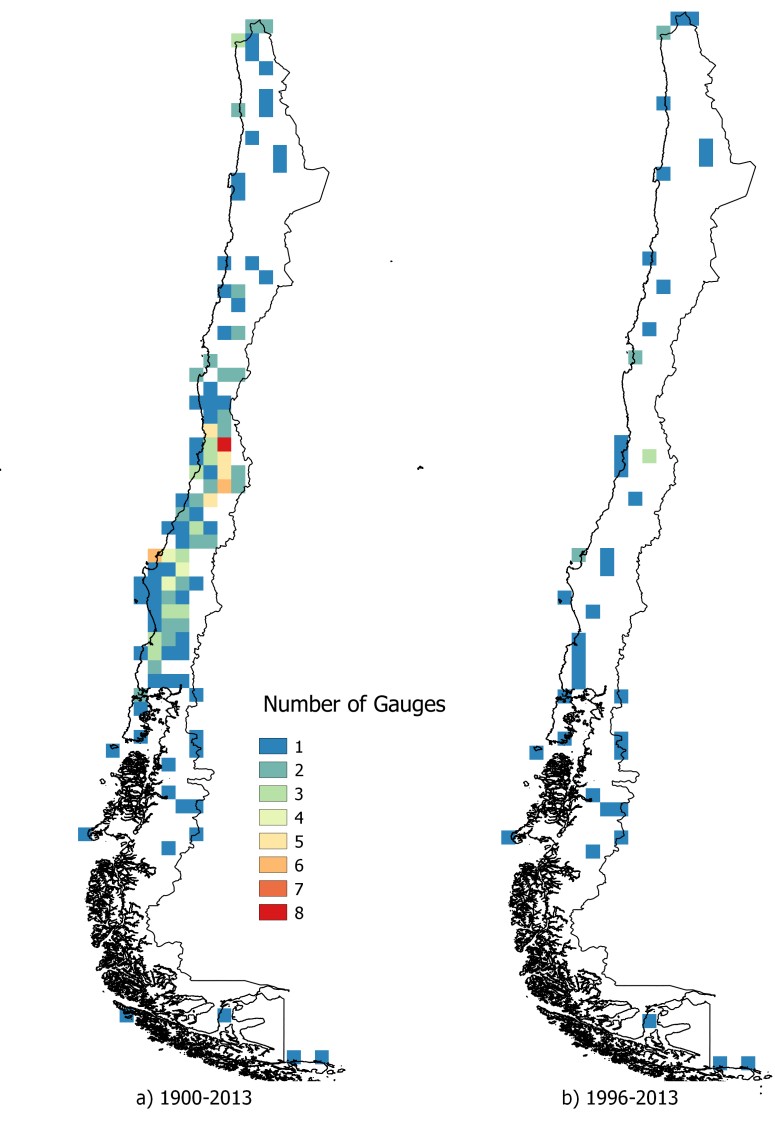

**Figure 3.** Spatial distribution of the maximum number of rain gauges used in Chile in GPCC over the full period and the maximum number of rain gauges used in the last two decades (Schamm et al., 2015).





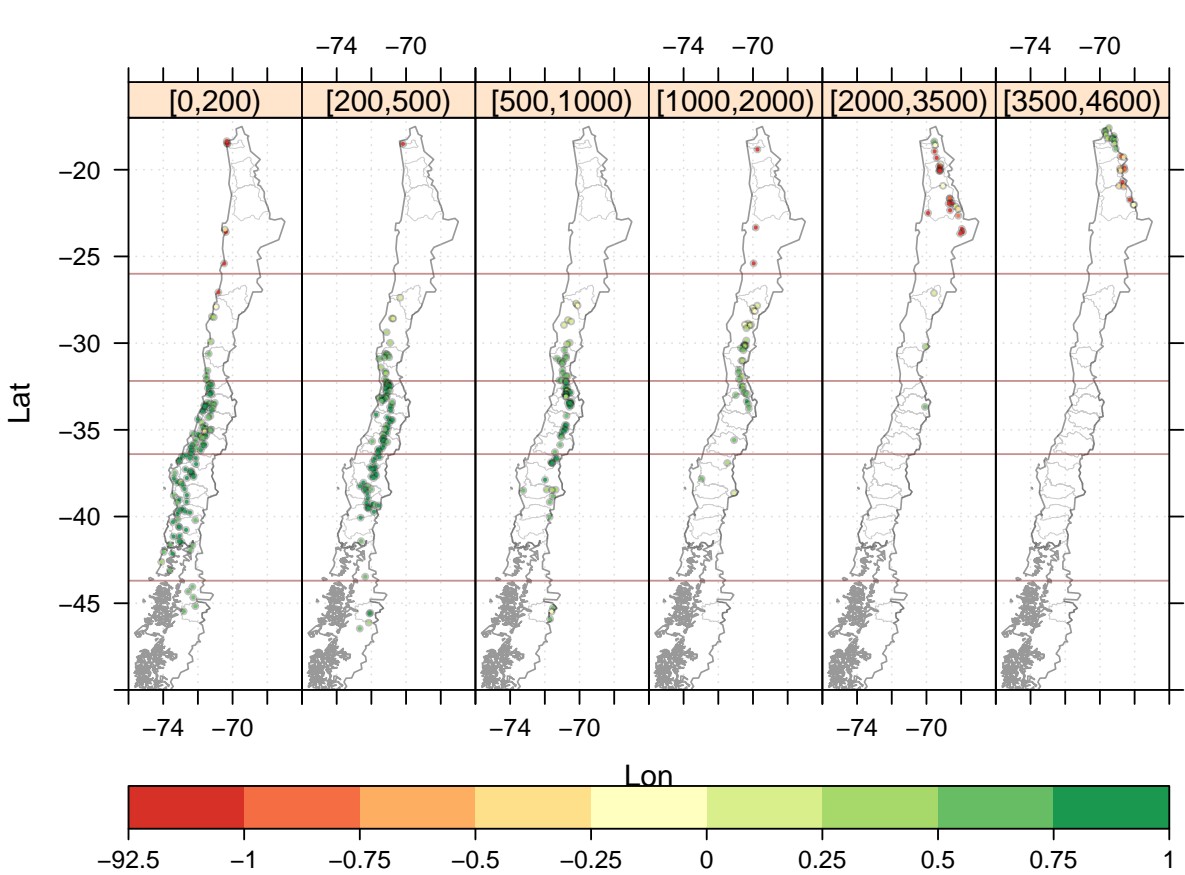

**Figure 4.** Modified Kling-Gupta efficiency ($KGE'$) between monthly satellite estimations (CHIRPS v2.0) and observations at the corresponding grid cell, for six different elevation zones: [0,200), [200,500), [500,1000), [1000,2000), [2000,3500), [3500,4560) m a.s.l. Colours for $KGE'$ range from intense red to dark green, representing very poor and optimum performance, respectively.





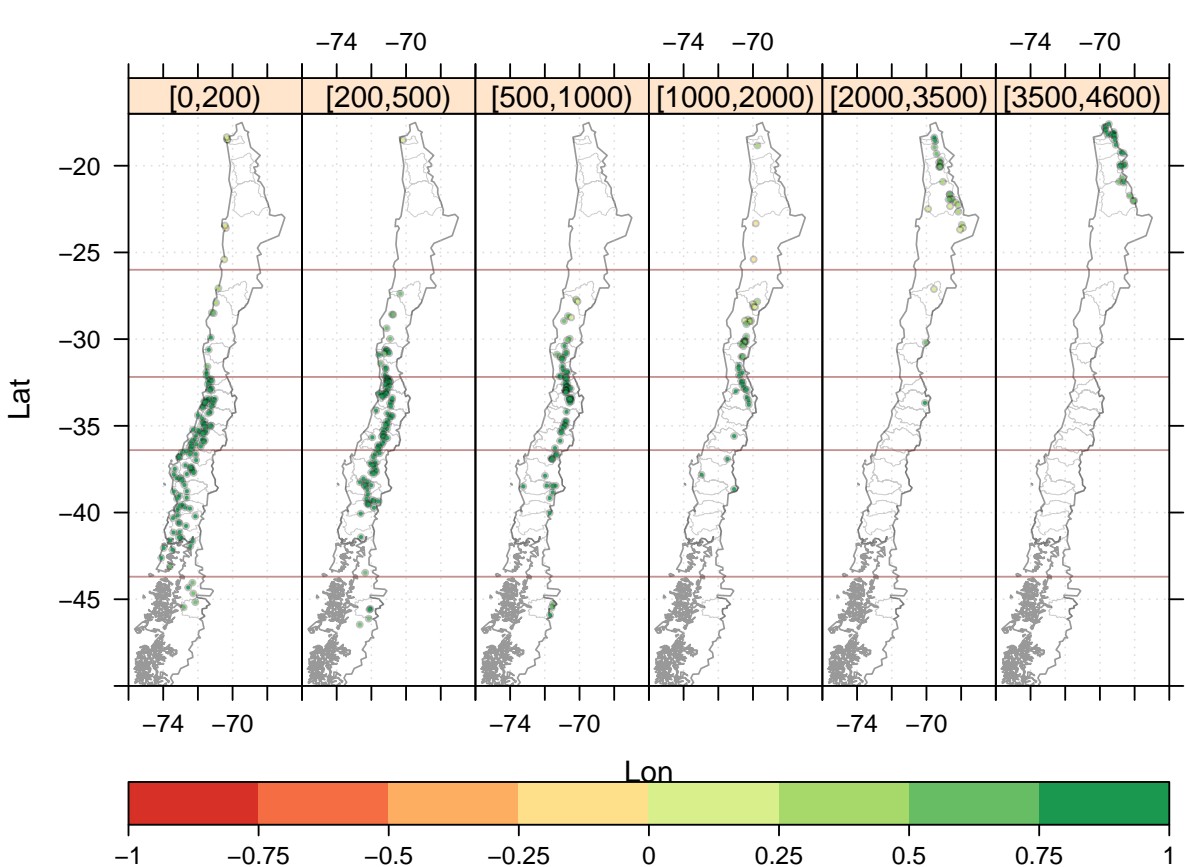

**Figure 5.** Pearson product-moment correlation coefficient ($r$) between monthly satellite estimations (CHIRPS v2.0) and the observations at the corresponding grid cell, for six different elevation zones: [0,200), [200,500), [500,1000), [1000,2000), [2000,3500), [3500,4560) m a.s.l. Colours for the $r$ performance range from intense red representing very poor linear correlation to intense green indicating a perfect positive linear correlation.





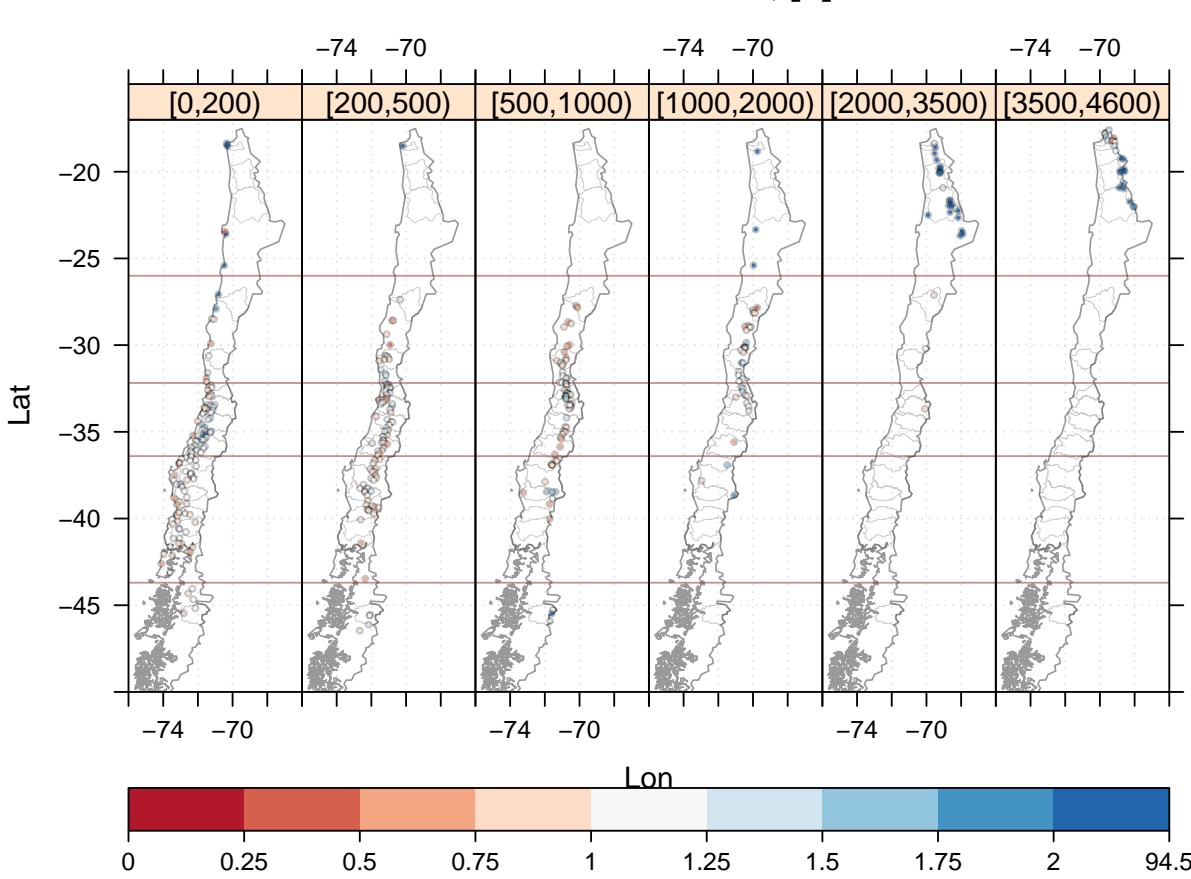

**Figure 6.** Bias ratio ($\beta$) between monthly satellite estimations (CHIRPS v2.0) and the observations at the corresponding grid cell, for six different elevation zones: [0,200), [200,500), [500,1000), [1000,2000), [2000,3500), [3500,4560) m a.s.l. Colours for $\beta$ range from intense red to dark blue representing a large under and overestimation of the observed amount of precipitation, respectively.





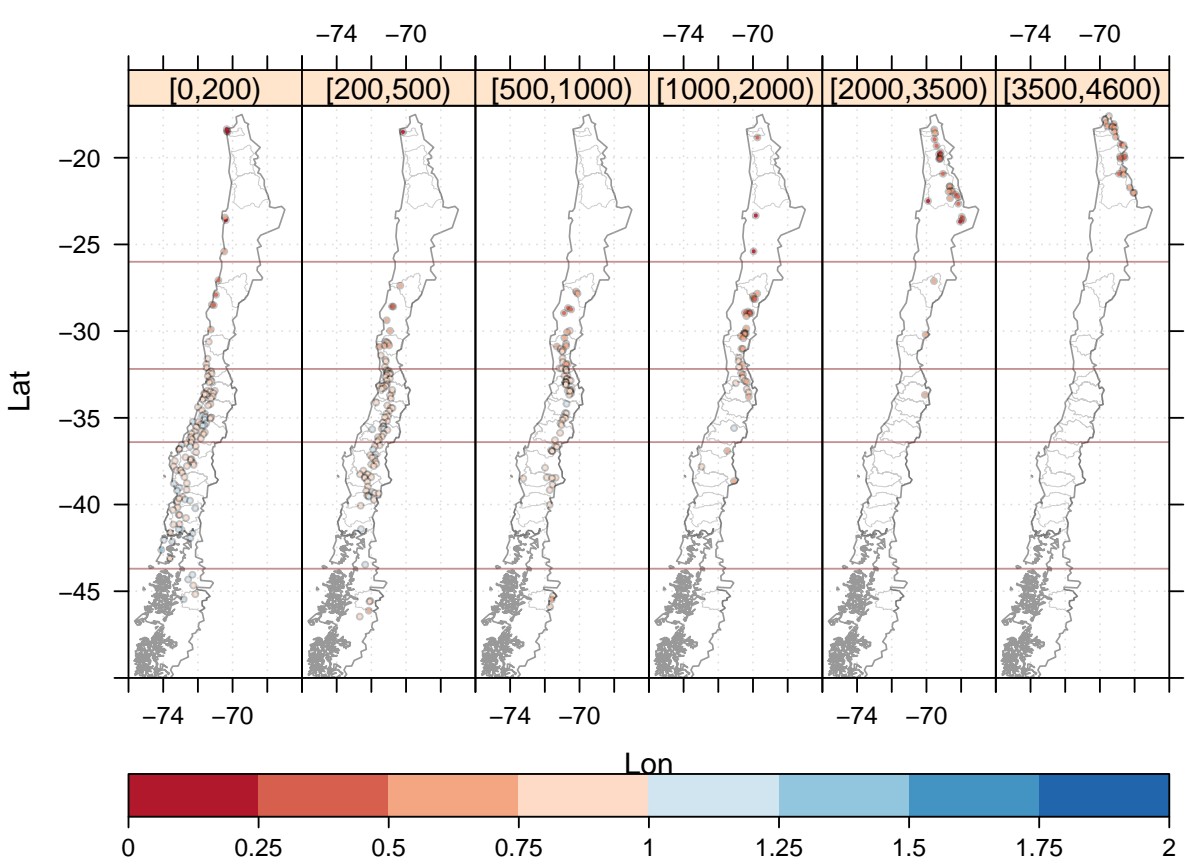

**Figure 7.** Variability ratio ($\gamma$) between monthly satellite estimations (CHIRPS v2.0) and the observations at the corresponding grid cell, for six different elevation zones: [0,200), [200,500), [500,1000), [1000,2000), [2000,3500), [3500,4560) m a.s.l. Colours for $\gamma$ range from intense red to dark blue representing a large under and overestimation of the observed variability of precipitation, respectively.





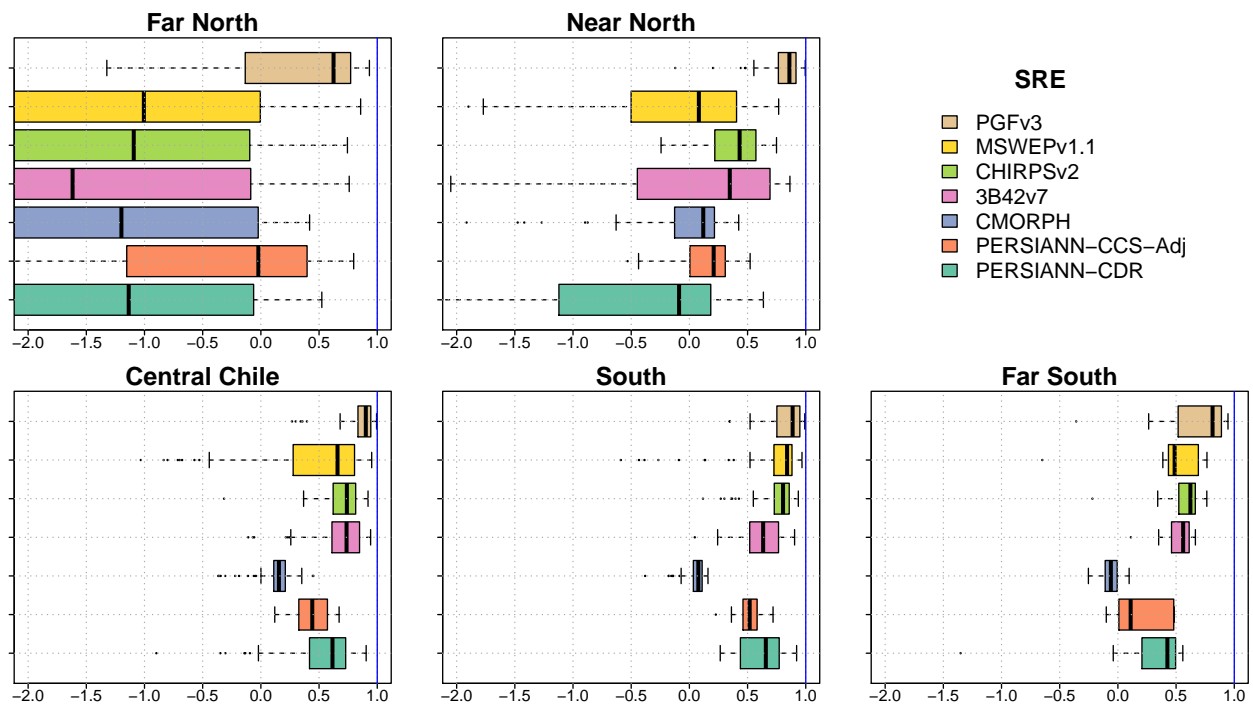

**Figure 8.** Modified Kling-Gupta efficiency (*KGE'*) between different monthly satellite estimations and observations at the corresponding grid cell, for five different macro-climatic zones: Far North, Near North, Centre, South and Far South. The vertical blue line indicates the optimum value for $KGE'$.





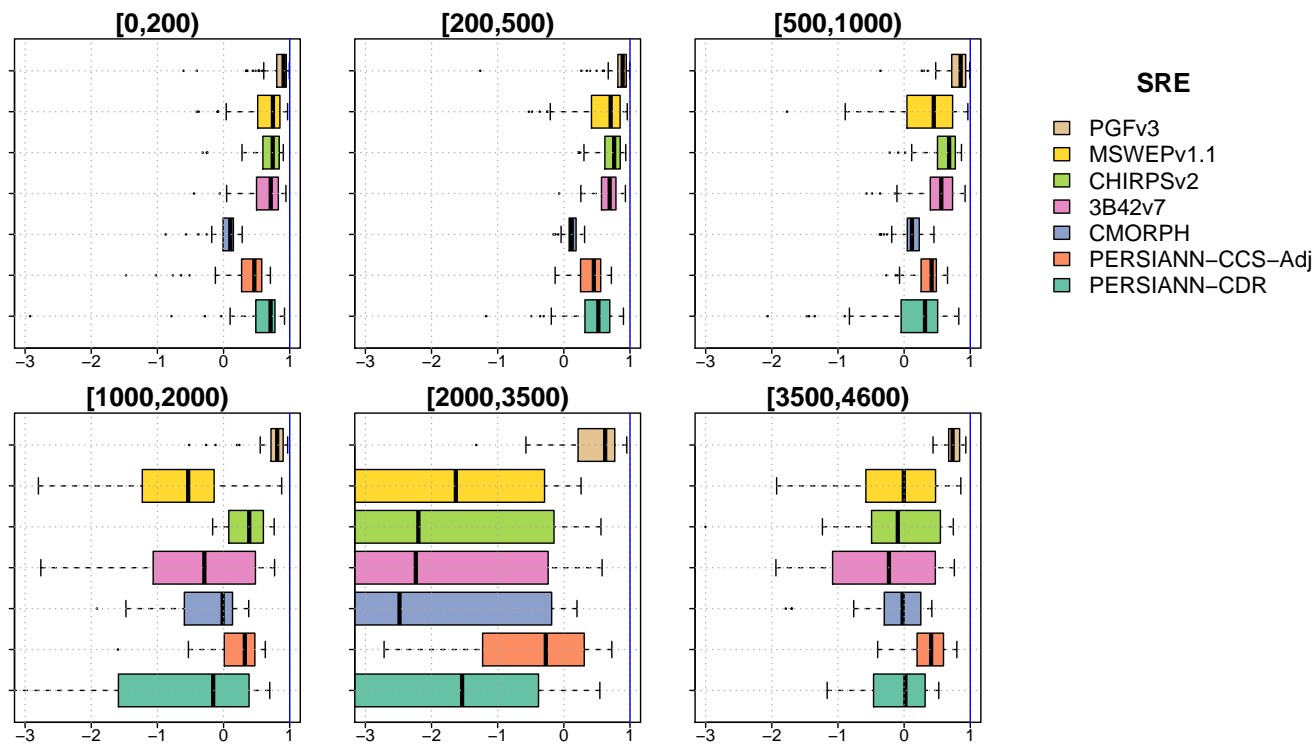

**Figure 9.** Modified Kling-Gupta efficiency (*KGE'*) between different monthly satellite estimations and observations at the corresponding grid cell, for six different elevation zones: [0,200), [200,500), [500,1000), [1000,2000), [2000,3500), [3500,4560) m a.s.l. The vertical blue line indicates the optimum value for $KGE'$.





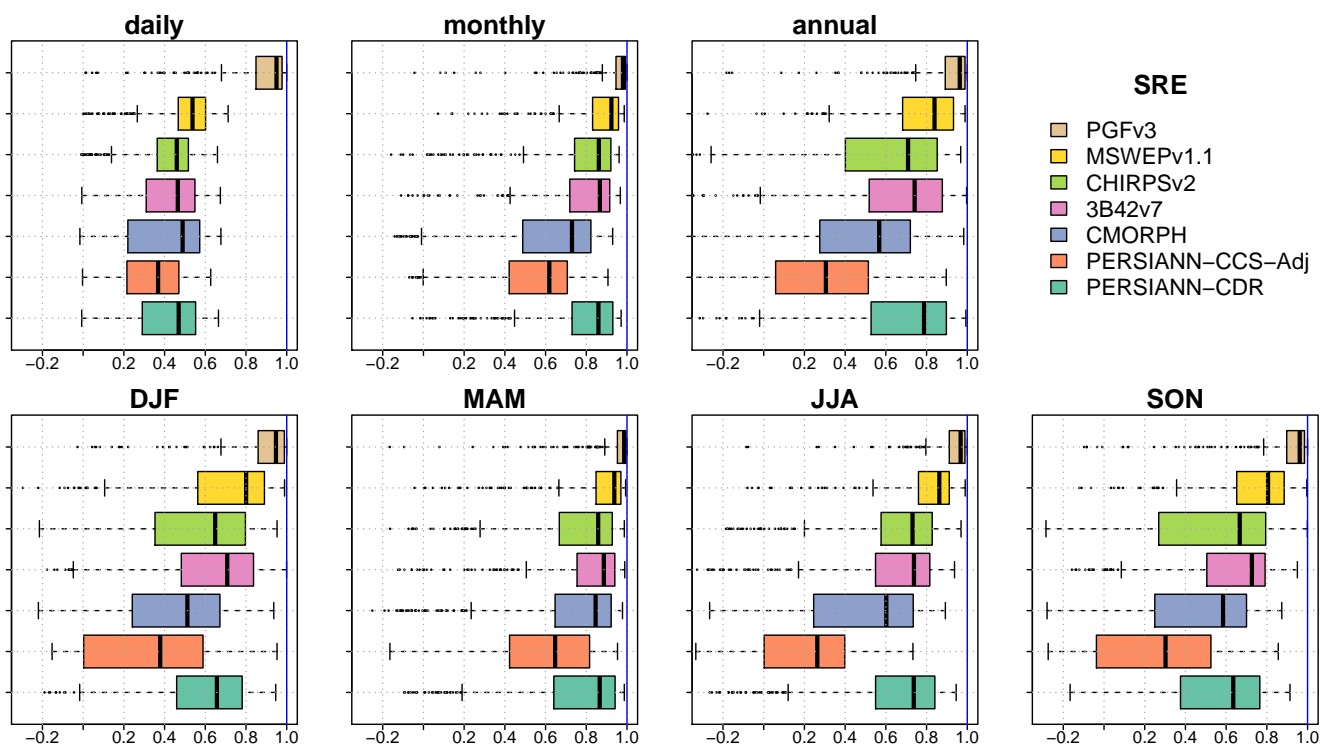

**Figure 10.** Pearson product-moment correlation coefficient ($r$) between different satellite estimations and the observations at the corresponding grid cell, for six different temporal scales. From left to right and up to bottom: daily, monthly, annual, summer (DJF), autumn (MAM), winter (JJA), and spring (SON). The vertical blue line indicates the optimum value for $r$.





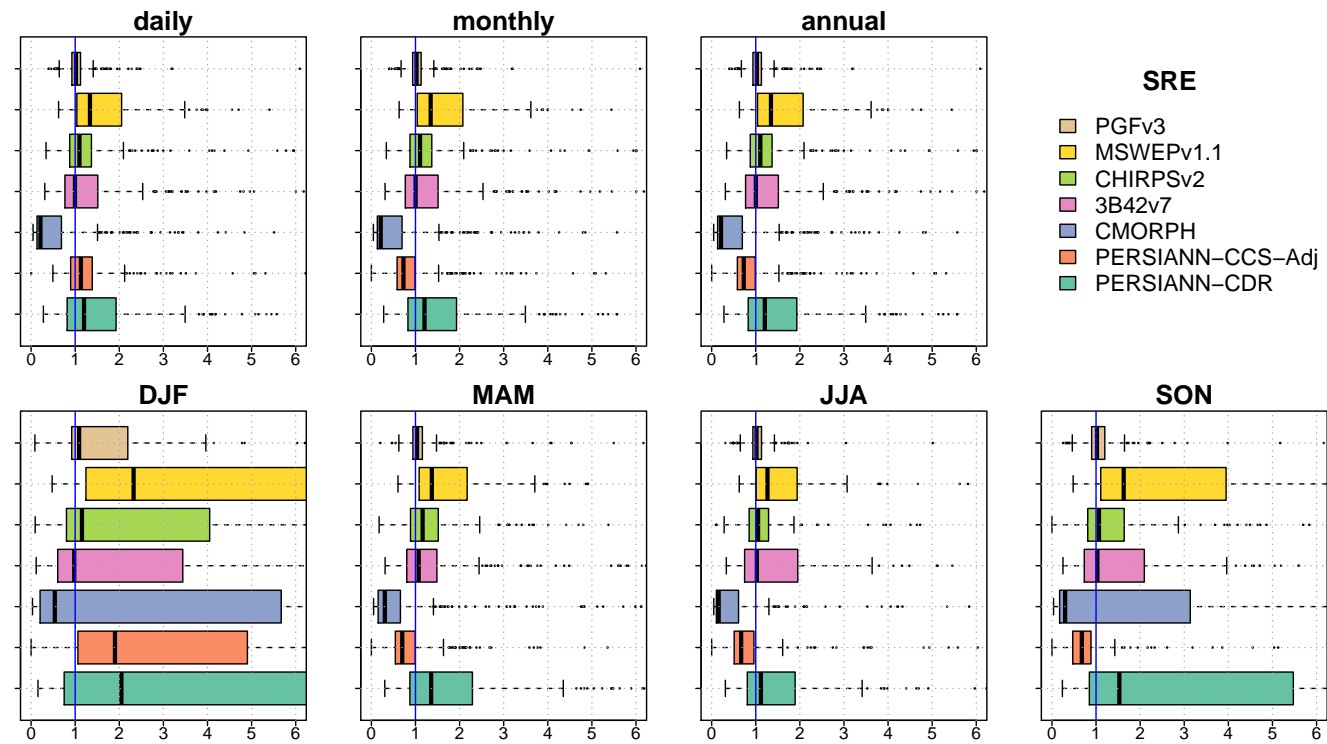

**Figure 11.** Bias ratio of the modified Kling-Gupta efficiency ($\beta$) between different satellite estimations and the observations at the corresponding grid cell, for six different temporal scales. From left to right and up to bottom: daily, monthly, annual, summer (DJF), autumn (MAM), winter (JJA), and spring (SON). The vertical blue line indicates the optimum value for $\beta$.





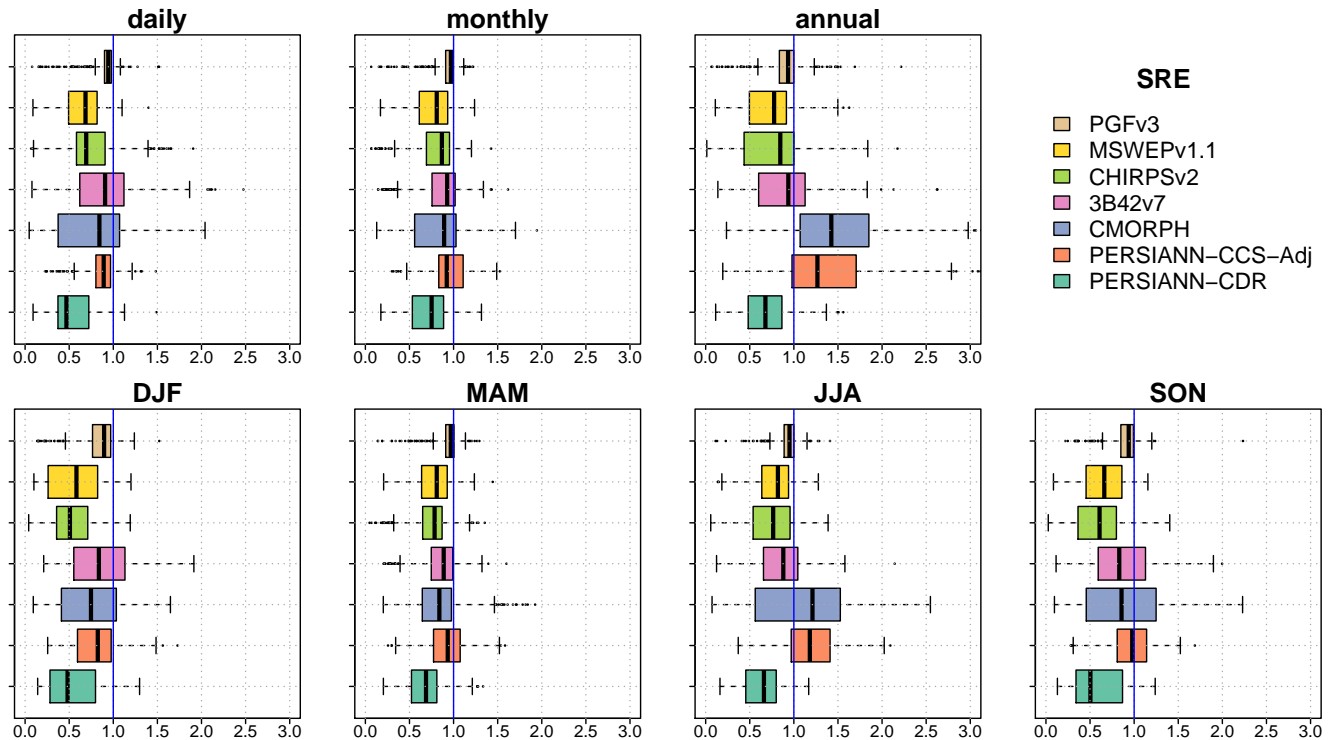

**Figure 12.** Variability ratio of the modified Kling-Gupta efficiency ($\gamma$) between different satellite estimations and the observations at the corresponding grid cell, for six different temporal scales. From left to right and up to bottom: daily, monthly, annual, summer (DJF), autumn (MAM), winter (JJA), and spring (SON). The vertical blue line indicates the optimum value for $\gamma$.





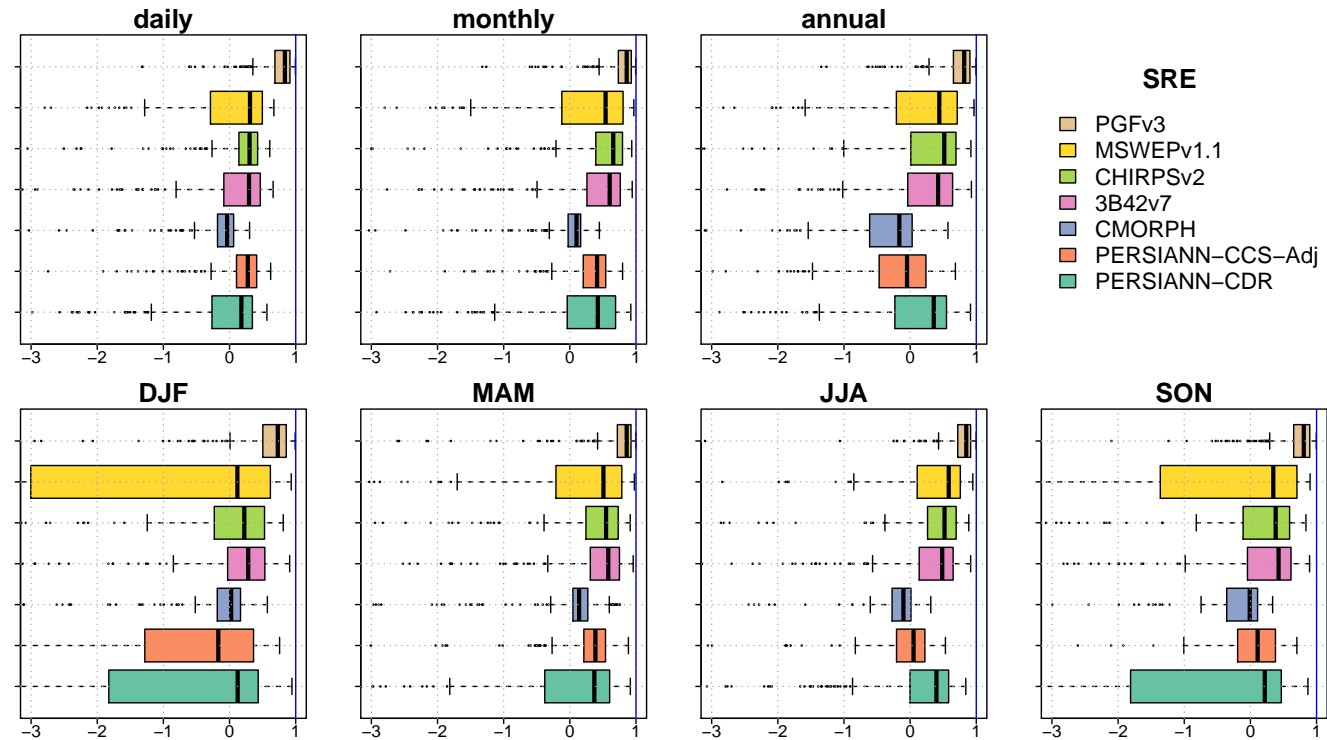

**Figure 13.** Modified Kling-Gupta efficiency (*KGE'*) between different satellite estimations and the observations at the corresponding grid cell, for six different temporal scales. From left to right and up to bottom: daily, monthly, annual, summer (DJF), autumn (MAM), winter (JJA), and spring (SON). The vertical blue line indicates the optimum value for $KGE'$.





**Figure 14.** Comparison of monthly and annual time series of precipitation, as estimated by the seven SRE products used in this work against the observed values register at the corresponding raingauge. Panels a), b), c), d) and e) show an example for the Far North, Near North, Centre, South and Far South climate macrozones.



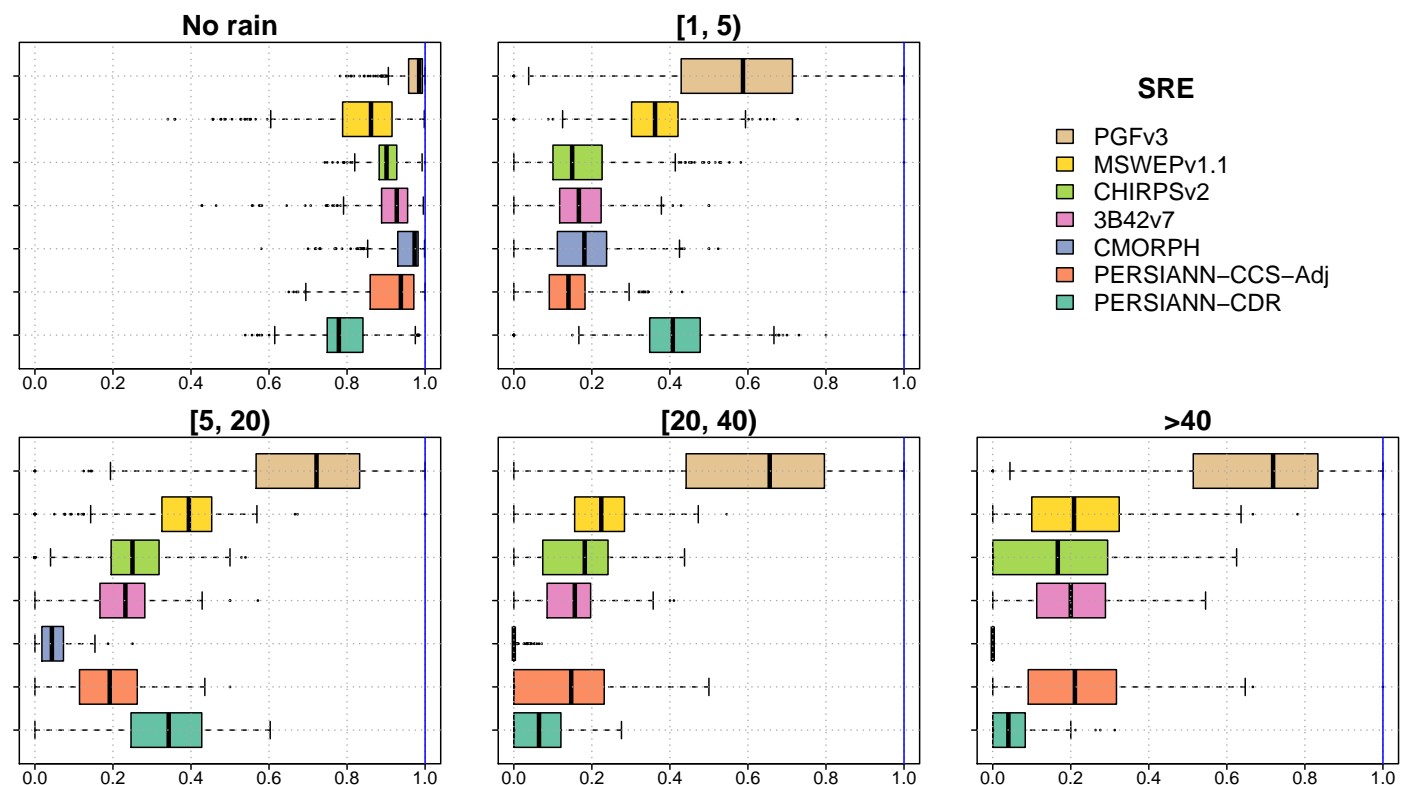

**Figure 15.** Probability of detection (*POD*) between different satellite estimations and the observations at the corresponding grid cell, for six different classes of daily precipitation. From left to right and up to bottom: [0,1), [1,5), [5,10), [10,50), $> 50$, [mm day$^{-1}$].





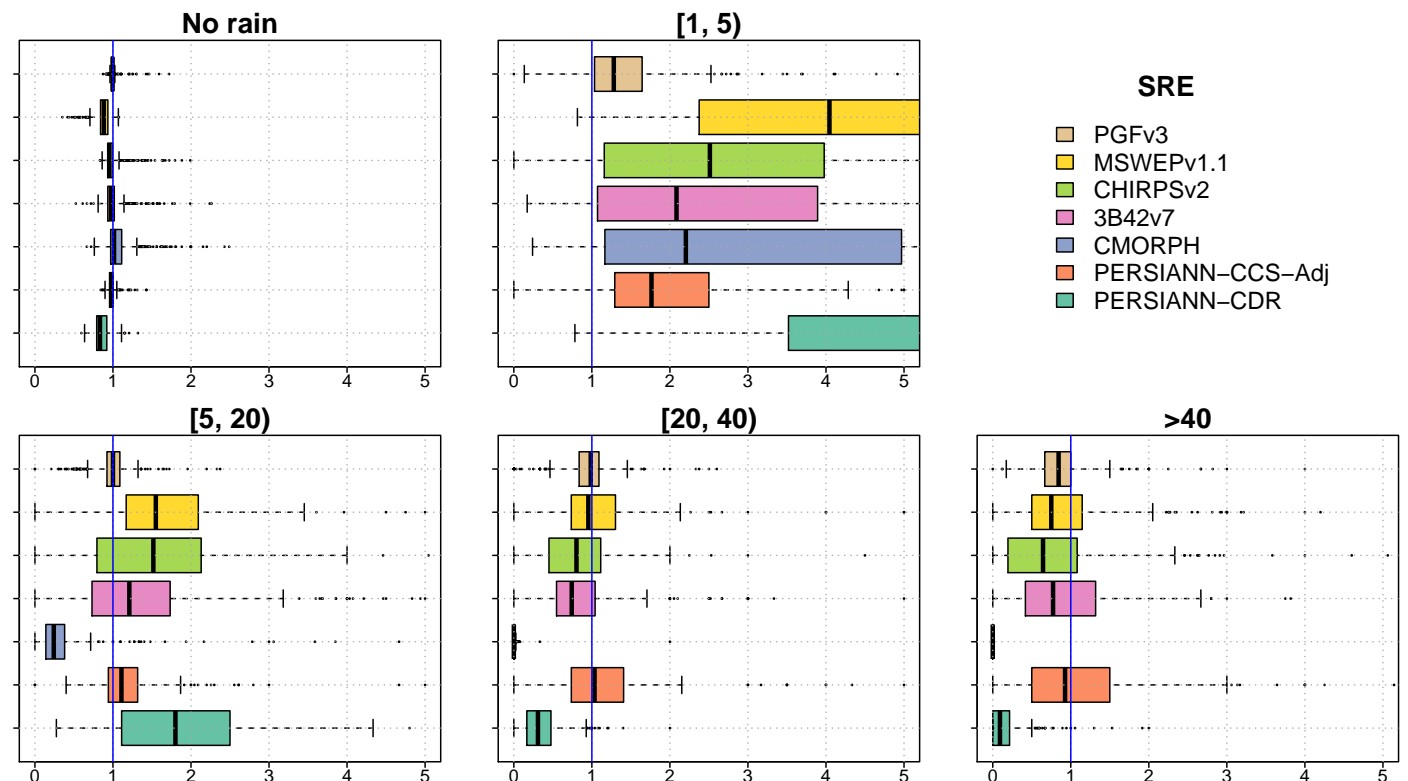

**Figure 16.** Frequency bias (*fBIAS*) between different satellite estimations and the observations at the corresponding grid cell, for six different classes of daily precipitation. From left to right and up to bottom: [0,1), [1,5), [5,10), [10,50), $> 50$, [mm day$^{-1}$].



**Table 1.** Summary of satellite-based products providing (sub)daily data at a quasi-global scale

| Data set | Full Name | Latitudinal Coverage | Spatial Resolution | Temporal Coverage | Temporal Resolutions | References |
|---|---|---|---|---|---|---|
| CMORPH | NOAA **C**limate Prediction Center (CPC) **MORPH**ing technique | 60°N-60°S | 0.25° | Dec-2002 - present | 3-hourly, daily | Joyce et al. 2004; CPC-NCEP-NWS-NOAA-USDC 2011 |
| PERSIANN-CDR | PERSIANN **C**limate **D**ata **R**ecord, Version 1 Revision 1 | 60°N-60°S | 0.25° | Jan-1983 - present | daily | Sorooshian et al. 2014; Ashouri et al. 2015 |
| PERSIANN-CCS-adj | **P**recipitation **E**stimation from **R**emotely **S**ensed **I**nformation using **A**rtificial **N**eural **N**etworks | 50°N-50°S | 0.04° | Jan-2003 - present | daily | Yang et al. 2016; Hong et al. 2004 |
| TMPA 3B42-V7 | **T**RMM **M**ulti-satellite **P**recipitation **A**nalysis research product 3B42 Version 7 | 50°N-50°S | 0.25° | Jan-1998 - present | 3-hourly, daily | Huffman et al. 2007 |
| CHIRPS V2.0 | **C**limate **H**azards group **I**nf**r**ared **P**recipitation with **S**tations Version 2.0 | 50°N-50°S | 0.05° | Jan-1981 - present | daily, pentadal, monthly | Funk et al. 2015 |
| MSWEP v1.1 | **M**ulti-**S**ource **W**eighted-**E**nsemble **P**recipitation Version 1.1 | 90°N-90°S | 0.25° | Jan-1979 Dec-2014 | daily | Beck et al. 2016 |
| PGF v3 | **P**rinceton University **G**lobal Meteorological **F**orcing Version 3 | 17°S-57°S | 0.25° | Jan-1979 Dec-2010 | daily | Peng et al. 2016; Sheffield et al. 2006 |

**Table 2.** Classification of rainfall events based on its daily intensity $i$. Modified for daily values from World Meteorological Organization (2008).

| Rainfall event | Intensity ($i$), [mm d$^{-1}$] |
|---|---|
| No rain | $[0 , 1)$ |
| Light rain | $[1 , 5)$ |
| Moderate rain | $[5 , 20)$ |
| Heavy rain | $[20 , 40)$ |
| Violent rain | $\geq 40$ |

**Table 3.** Contingency table used to compute the categorical performance indices for each rainfall event shown in Table 2.

| Satellite-product | Observed rainfall | | |
|---|---|---|---|
| | Yes | No | **Total** |
| Yes | Hit ($H$) | False Alarm ($FA$) | $H + FA$ |
| No | Miss($M$) | Correct Negative ($CN$) | $M + CN$ |
| **Total** | $H + M$ | $FA + CN$ | $Ne$ |