# Peer review of "Temporal and spatial evaluation of satellite-based rainfall estimates across the complex topographical and climatic gradients of Chile"

_Hydrology and Earth System Sciences, 2016_

## Referee Comment (RC1) · V. Maggioni (Referee) · 29 Sep 2016

1. Although the literature is saturated with articles that validate satellite precipitation products using some ground-based reference, I always welcome this kind of studies, since the use of these products is still limited by the uncertainty associated with them. Moreover, the authors used a complete array of statistics (both continuous and categorical) and focused on a large study area that has not been previously investigated and that presents challenges in satellite precipitation estimation due to its complex terrain. I believe this work addresses issues that are relevant to the HESS readership. The manuscript is well presented and well written. 2. I would encourage the authors to review an article that I recently published in JHM: Maggioni, V., Meyers, P.C. and Robinson, M.D., 2016. A Review of Merged High-Resolution Satellite Precipitation Product Accuracy during the Tropical Rainfall Measuring Mission (TRMM) Era. Journal of Hydrometeorology, 17(4), pp.1101-1117. In this article, the authors may find studies that have not cited but that may be relevant to compare their findings with previous work. I would also like to invite the authors to join the International Precipitation Working Group (IPWG), which could largely benefit from their insights on the validation of SRE over mountainous regions in South America. 3. It is my understanding that the authors used the TMPA 3B42 research version and not the real-time in their analysis. Can this be clarified in the text? 4. I would add the information that all SREs were rescaled to a common 25 km grid in the abstract. 5. Table 1 shows that the spatial resolution of CMORPH is 0.25deg, but this product is also available on an 8km grid, as stated in the text. 6. I appreciate the effort of considering several statistical metrics to evaluate the SREs. However, can the author discuss whether all these metrics are necessary to assess the performance of these products? Hossain and Huffman (2008) came up with a list of error metrics for evaluating SREs for hydrological applications. How do the statics introduced by the authors compare to that list?

---

## Referee Comment (RC2) · Anonymous Referee #2 · 11 Oct 2016

The Authors provided a comprehensive analysis of 7 satellite-based rainfall products over Chile. The analysis was carried on by considering several continuous and categorical scores, taking into account different time scales (daily, monthly, annual and seasonal). Moreover, the assessment was carried on by considering different climatic zones and altitude ranges. The rainfall products were compared with observed rainfall obtained from 366 station over the Chilean territory. Six out seven of the analyzed products used gauge data to calibrate and correct the rainfall estimates. The paper provided useful insights on the quality of satellite-based rainfall estimates over the complex study area. The paper is well written and clear, but I have some comments that I think should be addressed before publication. 1) The analysis was carried on over a 0.25° grid,
changing the spatial resolution of some satellite products. Could the Authors explain the regridding procedure? Did they average all the pixels within a 0.25° grid box? I think that some details should be added in the manuscript; 2) Do the Authors think that the regridding procedure have any impact in the satellite products performance?; 3) Why the Authors did not consider real-time products? 4) Figure 14 description is completely missing from the text. 5) Do the Authors have an idea for the huge overestimation in the Far North region?

Minor comments: 1) P1, L16: I think that autumn should be changed in spring; 2) P5, L32: Please add the GPCC reference; 3) P6, L15: Please add some reference for the CMORPH European study; 4) P7, L20: It should be 3B43-V7; 5) P12, L1-2: I found the same sentence three times. Please check the text 6) Section 5 and 6: Please use the same name for the satellite products throughout the manuscript. 7) P41, P42: Please check the captions: the last two rainfall classes should be [10,40) and >40; 8) Table 1: The spatial resolution of CMORPH should be 0.08°.

---

## Referee Comment (RC3) · CM Massari (Referee) · 18 Oct 2016

**GENERAL COMMENTS**

This study carries out a validation study of seven state-of-the-art SRE products (TMPA 3B42v7, CHIRPSv2, CMORPH, PERSIANN-CDR, PERSIAN-CCS-adj, MSWEPv1.1 and PGFv3) over the complex topography and diverse climatic gradients of Chile. Different temporal scales (daily, monthly, seasonal, annual) are used in a point to-pixel comparison between precipitation time series measured at 366 stations (from sea level to 4600 m a.s.l. in the Andean Plateau) and the corresponding grid cell of each SRE. Different continuous and categorical scores are used to assess the ability of each SRE to correctly estimate precipitation amounts and intensities. Despite literature is plenty

of studies about the assessment of the performance of SRE, additional efforts in this direction is very welcome since:

- new products and new releases are produced every year which are not still validated (I would like to thank the authors to use also new datasets like MSWEP).

- it is not clear what are the best performance scores to be used and which is the effect of the benchmark on the final results.

- Only by validation studies we can understand the future directions in the improvement of sensors and retrieval algorithms.

For the above reasons I think the paper is of interest of the HESS reader. On the whole, I found the paper well written and organized. It contains a lot of material which however has to be better synthetized in some key figures. For this, I suggest the authors to make an effort for presenting less figures or finding alternative plots or tables able to synthetize the results. I also have some major concerns (but nothing that impedes the paper publications) that the authors should address before the paper can be considered publishable on HESS.

1. The first is the discussion of the accuracy and the density of the benchmark dataset. After divided the area in different zones (latitude and elevation) the authors come up with a different number or rain gauges for each zone for each elevation. Since performance scores can be affected from the network density I suggest to report also the number of gauges used for each latitude and altitude zone and discuss more potential effects on the results.

2. Higher elevations in winter may be affected by the problem of snow which can be missed by rain gauges. Can the authors can add a comment on that?

3. It is not clear the rationale behind the choice of the SRE products. Some of them are gauge adjusted while some others not (e.g. CMORPH) despite a gauge corrected version of this product exists. I expect some comments on that.

4. In the outlook section (final section) it reads "Despite continuous improvements of most SRE products, a site-specific calibration is still recommended before any use in hydrological studies. This was evidenced by the PGFv3 dataset that showed a better performance compared to the other products due to the statistical merging of local precipitation observations. This highlights the need for adequate station observation networks to complement the SRE products, due to the dependency of SRE product quality on this ground-truthing." Here, it is taken for granted that the present rain gauge network provides the best performance in hydrological modelling, and since PGFv3 compares well with it then it will provide the best hydrological simulation. However, there is no evidence (by a hydrological simulation) that this is true and even if can accept that this can be assumed reasonable where the rain gauge density is high it cannot be generalized everywhere, especially because PGF uses the same data of the rainfall network. I think this can make believe the reader that gauge data are the truth while they represent just a good (not in all cases) estimate of the truth rainfall. For instance, if the gauge density is too low the spatial representativeness of the rainfall might be seriously affected. I suggest to discuss more this statement or provide a proof able to demonstrate it.

Below I provide ADDITIONAL COMMENTS in order of appearance in the manuscript also indicating their relevance.

Pag. 2. Lines 24-29. MINOR. I think it is not relevant how other people called satellite precipitation estimates.

Pag 4. Lines 10. MODERATE. "The present study attempts to exhaustively evaluate -for the fist time- the suitability of seven state-of-the-art SRE products or hydrological application in this data-scarce and complex mountainous region". There is no hydrological validation of the SRE in the paper. Please consider revising.

Pag. 13 MODERATE. Figure 14 is not cited neither discussed in the text.

Pag. 14 lines 28 MODERATE. Annual performance worse than daily ones. There could

be a dependence on the number of points (only 8, 2003-2010 for the yearly analysis) and thus a scarce representativeness on the performance of the products? Or it is an effect of the systematic bias a shorter time scale which then is amplified at annual scale. Can you add a comment on that?

Supplementary material is not used in the text and some performance scores are calculated but never cited in the text.

COMMENTS ON THE FIGURES

1. The number of figures in the paper and in the supplementary material is enormous. I suggest to merge figures 4, 5, 6 and -7 in one figure and better highlight the color of the dots since they are difficult to read (you can use a different colorbar).

2. Box plots are interesting but you can also consider to organize figures 10 to 13 in a Table. It will show the same concepts and will allow a better comparison between the different components of the KGE.

3. POD and fBIAS can be shown in a figure where in the x-axis you have the rainfall classes and on the y-axis you have POD and FBIAS. Then different curves will point to different products

---

## Author Comment (AC1) · 29 Nov 2016

Our reply to the comment of the referee was uploaded in the form of a PDF supplement.

Please also note the supplement to this comment:
http://www.hydrol-earth-syst-sci-discuss.net/hess-2016-453/hess-2016-453-AC1-supplement.pdf

---

## Author Comment (AC2) · 29 Nov 2016

Response to reviews on the manuscript **hess-2016-453**
"*Temporal and spatial evaluation of satellite-based rainfall estimates across the complex topographical and climatic gradients of Chile*" by Zambrano-Bigiarini, Nauditt, Birkel, Verbist and Ribbe 2016

November 29, 2016

We would like first to thank the constructive comments and suggestions raised by V. Maggioni, C. Massari and an anonymous referee (AR2). In the next sections we provide a detailed list with the responses to all the remarks pointed out by the ARs, hoping to clarify all the raised issues. We proceed by listing the reviewers' comments (**in bold text**) and our corresponding reply.

**V. Maggioni (Referee #1)**

**Comments**

**AR1-C1. Although the literature is saturated with articles that validate satellite precipitation products using some ground-based reference, I always welcome this kind of studies, since the use of these products is still limited by the uncertainty associated with them. Moreover, the authors used a complete array of statistics (both continuous and categorical) and focused on a large study area that has not been previously investigated and that presents challenges in satellite precipitation estimation due to its complex terrain. I believe this work addresses issues that are relevant to the HESS readership. The manuscript is well presented and well written.**

AR1-R1

We thank the referee #1 for her comments about our contribution.

**AR1-C2. I would encourage the authors to review an article that I recently published in JHM: Maggioni, V., Meyers, P.C. and Robinson, M.D., 2016. A Review of Merged High-Resolution Satellite Precipitation Product Accuracy during the Tropical Rainfall Measuring Mission (TRMM) Era. Journal of Hydrometeorology, 17(4), pp.1101-1117. In this article, the authors may find studies that have not cited but that may be relevant to compare their findings with previous work. I would also like to invite the authors to join the International Precipitation Working Group (IPWG), which could largely benefit from their insights on the validation of SRE over mountainous regions in South America.**

AR1-R2

We thank the referee for pointing out our attention to the article *Maggioni et al.* (2016). We reviewed it and compared our findings with those mentioned on it. Finally, we already wrote to the IPWG requesting to be considered as potential new members of the group.

**AR1-C3. It is my understanding that the authors used the TMPA 3B42 research version and not the real-time in their analysis. Can this be clarified in the text?**

AR1-R3

Corrected. In Page 7 we modified lines 20, 21, 29 and 30. In particular, we add the text "*In this work we only used the 3B42_daily product, which was accumulated to monthly, seasonal or annual values depending on the analysis*".

**AR1-C4. I would add the information that all SREs were rescaled to a common 25 km grid in the abstract**

AR1-R4

Done (line 10 in the abstract of the new manuscript).

**AR1-C5. Table 1 shows that the spatial resolution of CMORPH is 0.25deg, but this product is also available on an 8km grid, as stated in the text**

AR1-R5

CMORPH produces global precipitation analyses at very high spatial (8 km) and temporal (30 min) resolution starting from December 2002 until near-present. However, the data used in this work were 3-hr observations at a spatial resolution of 0.25° in NetCDF format. The 3-hr observations were accumulated to monthly, seasonal and annual totals depending on the analysis. There were two reasons for not using the very high-resolution product (8 km, 30 min): i) the webpage used to implement automatic downloading of data (*CPC-NCEP-NWS-NOAA-USDC*, 2011) only offered the possibility to define a spatial subset (i.e. the Chilean extent) for the 0.25x0.25° data, which saved hard disk space for storage, and ii) the final comparison among SREs was done using a 0.25x0.25° grid, so we preferred to use the

data directly from the provider instead of downloading them at 0.07x0.07° and then aggregating to a 0.25x0.25° grid for carrying out the comparisons. The authors are aware of the existence of the webpage `ftp://ftp.cpc.ncep.noaa.gov/precip/global_CMORPH/30min_8km` with data at the very high spatial and temporal resolution. However, it was not possible for us to obtain data files for any period before 2015. We will add a clarifying text in page 6, lines 17 and 18, and will add also the 0.07° resolution in Table 1 of the new manuscript.

**AR1-C6. I appreciate the effort of considering several statistical metrics to evaluate the SREs. However, can the author discuss whether all these metrics are necessary to assess the performance of these products? Hossain and Huffman (2008) came up with a list of error metrics for evaluating SREs for hydrological applications. How do the statics introduced by the authors compare to that list?**

AR1-R6
Done. We thank the referee for pointing our attention to the article of *Hossain and Huffman* (2008). First, we agree with *Hossain and Huffman* 2008, hereafter HH08, in that "*there is probably not a unique way of representing error completely*". Similar to them, our work also intends to increase the commonly used metrics (e.g., correlation, bias, RMSE, etc) in order to provide a better assessment of satellite-based precipitation products over the Chilean territory at hydrologically relevant temporal and spatial scales.

HH08 defined the following error metrics:

1. Probability of rain detection ($POD_{rain}$), as a function of rainfall magnitude

2. Probability of no-rain detection ($POD_{norain}$)

3. First- and second-order moments of the probability distribution during false alarms

4. Correlation lengths for the detection of rain ($CL_{rain}$)

5. Correlation lengths for the detection of no rain ($CL_{norain}$)

6. Conditional systematic retrieval error or mean-field bias (when reference $rain > 0$)

7. Conditional random retrieval error or error variance

8. Correlation length for the retrieval error ($CL_{ret}$) (conditional, when $rain > 0$)

9. Lag one autocorrelation of the mean field bias.

Their work was focused on the use of the previous error metrics for simulating stochastic realizations of satellite rainfall with realistic space-time covariance structures. On the other hand, our work was focused on using error metrics that easily convey quantitative information about the overall skill of different SREs in representing the "ground-truth" (KGE and its three components) as well as the skill of each SRE in representing different rainfall intensities ($POD$, $fBias$, $FAR$, $ETS$).

Notwithstanding both HH08 and our work computed the probability of detection for different rainfall intensities ($POD_{r}ain$), there is a key difference between them:

$POD_{rain} = Prob\{R_{SAT} > 0 | R_{REF} > t_0\}$ , HH08
$POD_{rain} = Prob\{R_{SAT} \in [t_0, t_1] | R_{REF} \in [t_0, t_1]\}$, our work

so, we are not only interested in knowing if a SRE identified that "it was raining" when the reference observation was larger than a user-defined threshold $t_0$, but we are interested in knowing if both the reference rainfall value and the satellite estimate were comparable in its magnitude, i.e., both of them were in the same user-defined interval $[t_0, t_1]$. To summarise the previous point, we agree with HH08 that $POD_{rain}$ and $POD_{norain}$ are important measures to quantify the ability of a SRE to detect the magnitude of the rainfall rate, but we propose to use the same interval of rainfall rates for the reference and the SRE values in order to classify a "Hit".

Finally, given the increasing use of the Kling-Gupta efficiency ($KGE$) by the hydrological community since 2009 (e.g. *Thiemig et al.*, 2013; *Coron et al.*, 2013; *Pechlivanidis et al.*, 2014; *Crochemore et al.*, 2015; *Bisselink et al.*, 2016) we think that using the KGE value and its three components (correlation, bias and variability) evaluated at daily, monthly, seasonal and annual time scales is easier to interpret and use by hydrologists than measures 3-9 analysed by HH08.

As a final summary, we recommend the use of $KGE$ and its three components $(r, \beta, \gamma)$ as continuous indices of model performance at different temporal scales, and the use of $POD$ and $fBias$ as categorical indices of model performance for different rainfall intensities (after evaluating $PC$, $POD$, $fBias$, $FAR$ and $ETS$).

**Anonymous Reviewer #2**

The Authors provided a comprehensive analysis of **7 satellite-based rainfall products over Chile**. The analysis was carried on by considering several continuous and categorical scores, taking into account different time scales (daily, monthly, annual and seasonal). Moreover, the assessment was carried on by considering different climatic zones and altitude ranges. The rainfall products were compared with observed rainfall obtained from 366 station over the Chilean territory. Six out seven of the analyzed products used gauge data to calibrate and correct the rainfall estimates. The paper provided useful insights on the quality of satellite-based rainfall estimates over the complex study area. The paper is well written and clear, but I have some comments that I think should be addressed before publication.

**Comments**

**AR2-C1. The analysis was carried on over a 0.25° grid, changing the spatial resolution of some satellite products. Could the Authors explain the regridding procedure? Did they average all the pixels within a 0.25° grid box? I think that some details should be added in the manuscript.**

AR2-R1
The upscaling procedure applied in this work consisted in transferring values from the high resolution raster cells to each one of the 0.25° grid cells, by using bi-linear interpolation as implemented in the `resample` function of the `raster` R package (*Hijmans*, 2016). This upscaling was carried out in order to enable a consistent point-to-pixel comparison between SREs and ground-based data, which is widely used in comparative studies (e.g. *Chen et al.*, 2016; *Thiemig et al.*, 2012; *Novella and Thiaw*, 2010; *Dinku et al.*, 2007). The aforementioned details about the regridding procedure will bee added in page 10, lines 7-10 of the new manuscript.

**AR2-C2. Do the Authors think that the regridding procedure have any impact in the satellite products performance?**

AR2-R2
In our opinion, to ensure that the regridding procedure has no effect on the final results of the comparison a specific study would be required, by testing different regridding procedures (e.g., bilinear, nearest neighbor, inverse distance, spline, binning, spectral, triangulation) on different time scales (3-hr, daily, monthly, seasonal, annual). However, based on the discussion on `https://climatedataguide.ucar.edu/climate-data-tools-and-analysis/regridding-overview`, we are confident about discarding effects of the regridding procedure on our results, because it was applied only to the daily values of PERSIANN-CCS-adj and CHRIPSv2 products, which are assumed to be smoothly varying variables within each 0.25° grid cell, and therefore the bi-linear interpolation would be an adequate interpolation technique. A short mention to the possible impact of the regridding technique on our results will be added in page 10, lines 10-12 of the new manuscript.

**AR2-C3. Why the Authors did not consider real-time products?**

AR2-R3
As the most suitable SRE product found in our research will be used for hydrological modelling in selected basins, using both historical and future climate change scenarios, we focused only on SREs with long data records and good quality data over the Chilean territory, without giving any priority to low-latency (near)real-time products. In addition, comparability of (near)real-time products and station-based calibrated research products is low (*Habib et al.*, 2009; *Huffman et al.*, 2007). PERSIANN-CCS-adj is the only product that provides both (near)real-time and historical estimates of precipitation (because it is especially developed for Chile), and that is the reason why it was included in our comparison study. A comparison of (near)real-time only products is very likely to be subject of future research of our team.

**AR2-C4. Figure 14 description is completely missing from the text.**

AR2-R4
Corrected. In the new version of the manuscript, one paragraph on Figure 14 will be added at the end of the results section:

"*Figure 14 compares the monthly and annual precipitation time series as estimated by the seven SRE products with the observed values registered by rain gauges at selected sites for the Far North, Near North, Centre, South and Far South climatic macrozones*".

In addition, one sentence on Figure 14 will also be added in the Discussion: "*Time series comparison between rain gauges and SRE data can be found in the supplementary material (Zambrano-Bigiarini et al., 2016) for each of the 366 selected stations described in Section 2.2.1*" (see also comparison of selected stations in Figure 14).

**AR2-C5. Do the Authors have an idea for the huge overestimation in the Far North region?**

AR2-R5
As the referee mentioned, SREs showed a large overestimation of precipitation values in the hyper-arid Far North (17.5-26.0°S). This can be attributed to the very low precipitation values in that area (some rainguges have annual values lower than 1 mm/year), where one single event not correctly identified can be responsible of 100% of over or underestimation. This overestimation has been observed as well in the scope of other studies (*Dinku et al.*, 2010; *Maggioni et al.*, 2016).

**Minor comments**

**AR2-mC1. P1, L16: I think that autumn should be changed in spring.**

AR2-mR1.
Corrected.

**AR2-mC2. P5, L32: Please add the GPCC reference.**

AR2-mR2
Added (P6, L5 of the new manuscript).

**AR2-mC3. P6, L15: Please add some reference for the CMORPH European study.**

AR2-mR3
Added (P6, L20 of the new manuscript).

**AR2-mC4. P7, L20: It should be 3B43-V7.**

AR2-mR4
See our reply AR1-R3.

**AR2-mC5. P12, L1-2: I found the same sentence three times. Please check the text.**

AR2-mR5
Corrected. The sentence "*Daily time series in all the 781 stations analysed in this work can be found in the supplementary material*" was mistakenly used four times in the text. Only the sentence in P6 lines 2 and 3 of the new manuscript will be kept.

**AR2-mC6. Section 5 and 6: Please use the same name for the satellite products throughout the manuscript.**

AR2-mR6
Done. The names used for the seven selected products were: CMORPH, CHIRPSv2, 3B42v7, MSWEPv1.1, PGFv3, PERSIANN-CDR, PERSIANN-CCS-adj.

**AR2-mC7. P41, P42: Please check the captions: the last two rainfall classes should be $[10, 40)$ and $> 40$**

AR2-mR7
Corrected.

**AR2-mC8. Table 1: The spatial resolution of CMORPH should be 0.08°**

AR2-mR8
Se our reply AR1-R5.

**C. Massari (Referee #3)**

This study carries out a validation study of seven state-of-the-art SRE products (TMPA 3B42v7, CHIRPSv2, CMORPH, PERSIANN-CDR, PERSIAN-CCS-adj, MSWEPv1.1 and PGFv3) over the complex topography and diverse climatic gradients of Chile. Different temporal scales (daily, monthly, seasonal, annual) are used in a point to-pixel comparison between precipitation time series measured at 366 stations (from sea level to 4600 m a.s.l. in the Andean Plateau) and the corresponding grid cell of each SRE. Different continuous and categorical scores are used to assess the ability of each SRE to correctly estimate precipitation amounts and intensities. Despite literature is plenty of studies about the assessment of the performance of SRE, additional efforts in this direction is very welcome since:

- new products and new releases are produced every year which are not still validated (I would like to thank the authors to use also new datasets like MSWEP).

- it is not clear what are the best performance scores to be used and which is the effect of the benchmark on the final results

- Only by validation studies we can understand the future directions in the improvement of sensors and retrieval algorithms

For the above reasons I think the paper is of interest of the HESS reader. On the whole, I found the paper well written and organized. It contains a lot of material which however has to be better synthetized in some key figures. For this, I suggest the authors to make an effort for presenting less figures or finding alternative plots or tables able to synthetize the results. I also have some major concerns (but nothing that impedes the paper publications) that the authors should address before the paper can be considered publishable on HESS.

**Comments**

**AR3-C1.** The first is the discussion of the accuracy and the density of the benchmark dataset. After divided the area in different zones (latitude and elevation) the authors come up with a different number or rain gauges for each zone for each elevation. Since performance scores can be affected from the network density I suggest to report also the number of gauges used for each latitude and altitude zone and discuss more potential effects on the results.

AR3-R1
We appreciate the comment of the referee. The number of gauges used for each latitude and altitude zone will be reported in the new version of the manuscript, with a discussion about the potential impacts on the results.

**AR3-C2.** Higher elevations in winter may be affected by the problem of snow which can be missed by rain gauges. Can the authors can add a comment on that?

AR3-R2
We agree with reviewer 3. As described in the introduction, uncertainties related to high elevation precipitation are extremely high as there are only very few climate stations above 1000 m a.s.l. As these few stations do not have any snow sensor, they cannot provide information on the snow water equivalent. Three stations with snow pillows were built up during the last three years, and measurements are being collected for future comparisons. Therefore, we expect to quantify these uncertainties by using SREs in hydrological modelling to close the water balance.

We propose to the referee the addition of the following sentence to the new manuscript (Page 1, lines 10-12):

"*Moreover, uncertainties related to precipitation in high-elevation areas can additionally be attributed to missing snow monitoring equipment (i.e., snow scale or pillow to determine the snow-water equivalent), reducing accuracy of precipitation measurements during the winter season.*"

**AR3-C3. It is not clear the rationale behind the choice of the SRE products. Some of them are gauge adjusted while some others not (e.g. CMORPH) despite a gauge corrected version of this product exists. I expect some comments on that.**

AR3-R3
See AR2-R3. A short comment on the rationale behind the selection of the SRE products will be added at the beginning of the Section 2.2.2 of the new manuscript.

**AR3-C4. In the outlook section (final section) it reads** *Despite continuous improvements of most SRE products, a site-specific calibration is still recommended before any use in hydrological studies. This was evidenced by the PGFv3 dataset that showed a better performance compared to the other products due to the statistical merging of local precipitation observations. This highlights the need for adequate station observation networks to complement the SRE products, due to the dependency of SRE product quality on this ground-truthing.* **Here, it is taken for granted that the present rain gauge network provides the best performance in hydrological modelling, and since PGFv3 compares well with it then it will provide the best hydrological simulation. However, there is no evidence (by a hydrological simulation) that this is true and even if can accept that this can be assumed reasonable where the rain gauge density is high it cannot be generalized everywhere, especially because PGF uses the same data of the rainfall network. I think this can make believe the reader that gauge data are the truth while they represent just a good (not in all cases) estimate of the truth rainfall. For instance, if the gauge density is too low the spatial representativeness of the rainfall might be seriously affected. I suggest to discuss more this statement or provide a proof able to demonstrate it.**

AR3-R4
We fully agree with the referee comment. Up to date there is no way to provide fully reliable estimates of areal catchment precipitation, especially with high spatial and temporal resolution. Spatial variability of rainfall is very high and cannot be exactly measured even by a very dense network of gauging instruments.

We propose to add a short Section 3.3 entitled "Uncertainties in the verification results" with the following text:

"*Rainfall measurements using standard gauges and different types of tipping-bucket and weighing gauges often cannot accurately determine precipitation in a certain location, because of the impact of wind, wetting losses, evaporation and other systematic error sources on rain, snow and precipitation intensity (Legates and DeLiberty, 1993; Sevruk et al., 2009). Therefore, there are large uncertainties inherent in the ground data used for verification (Scheel et al., 2011). A method to appropriately take into account errors in the ground observations to then quantify uncertainties in the verification results is still a challenge for the scientific community (Ebert, 2007). Therefore, validated SRE products should not replace gauge-based observations but provide complementary information (Scheel et al., 2011). The application of real-time SRE products in hydrological modelling will contribute to close such gaps in the water balance and to quantify the related uncertainties.*"

**Additional comments**

**AR3-aC1. Pag. 2. Lines 24-29. MINOR. I think it is not relevant how other people called satellite precipitation estimates.**

AR3-aR1
We partially agree with the referee. The individual name that each research group uses to call satellite precipitation estimates does not make any difference at the moment of evaluating their performance. However, due to the increasing interest in on SRE products, we prefer to provide a short summary of possible names found in literature as a quick guide to those beginners searching for validations or applications of different products.

**AR3-aC2. Pag 4. Lines 10. MODERATE.** *The present study attempts to exhaustively evaluate -for the fist time- the suitability of seven state-of-the-art SRE products for hydrological application in this data-scarce and complex mountainous region.* **There is no hydrological validation of the SRE in the paper. Please consider revising.**

AR3-aR2
Corrected. We propose to change the previous sentence by:

"*The present study attempts to exhaustively compare -for the fist time- seven state-of-the-art SRE products against ground-based measurements in this data-scarce and complex mountainous region, in order to provide guidelines about suitable SRE products for future hydrological applications*".

**AR3-aC3. Pag. 13 MODERATE. Figure 14 is not cited neither discussed in the text.**

AR3-aR3
Corrected. See AR2-R4.

**AR3-aC4. Pag. 14 lines 28 MODERATE. Annual performance worse than daily ones. There could be a dependence on the number of points (only 8, 2003-2010 for the yearly analysis) and thus a scarce representativeness on the performance of the products? Or it is an effect of the systematic bias a shorter time scale which then is amplified at annual scale. Can you add a comment on that?**

AR3-aR4
We propose to the referee to change the current text:

"*which very likely is due to the effect of monthly to annual temporal aggregation, where small differences between monthly SRE values and observations during a large number of months are then summed up when aggregated into a smaller number of annual values*"

by:

"*which was due to the amplification of small systematic biases at the daily time scale when aggregated into the annual scale, passing from 2922 days to only 8 annual values*".

**AR3-aC5. Supplementary material is not used in the text and some performance scores are calculated but never cited in the text.**

AR3-aR5
Corrected. Supplementary material will be briefly mentioned in selected sentences of the text as well as all the performance measures ($KGE$, $r$, $\alpha$, $\gamma$, $PC$, $POD$, $ETS$, $FAR$, $fBias$).

**Comments on the figures**

**AR3-fC1. The number of figures in the paper and in the supplementary material is enormous. I suggest to merge figures 4, 5, 6 and -7 in one figure and better highlight the colour of the dots since they are difficult to read (you can use a different colorbar).**

AR3-fR1
Figures 5, 6 and 7 will be moved to the Supplementary material, and only the current Figure 4 will be kept, because it summarises the previously mentioned three figures. Regarding the colorbar, the one used for the figures was selected following the *modified spectral scheme* suggested in the article `https://eos.org/features/the-end-of-the-rainbow-color-schemes-for-improved-data-graphics`. However, the authors will try to find a better combination of colours for the new manuscript. Finally, each one of the PDF files included in the supplementary material allow to make use of different levels of zoom, in order to clearly appreciate differences among point values.

**AR3-fC2. Box plots are interesting but you can also consider to organize figures 10 to 13 in a Table. It will show the same concepts and will allow a better comparison between the different components of the KGE.**

AR3-fR2
We started using tables and then moved into boxplots, mainly because of the very large amount of numbers produced in each table. In particular, each one of the boxplots presented in figures 10 to 13 summarises the empirical distribution of 366 values, so a table trying to depict a similar amount of information would contain 17934 rows (7 SRE x 366 stations x 7 temporal scales). Therefore, we prefer to keep the boxplots because they visually summarise the information that in a table would require much more space.

**AR3-fC3. POD and fBIAS can be shown in a figure where in the x-axis you have the rainfall classes and on the y-axis you have POD and FBIAS. Then different curves will point to different products.**

AR3-fR3

Thanks for the comment. We will implement two new figures (one for $POD$ and other for $fBias$) in the new version of the manuscript. However, considering the comment AR3-fC1, those figures will be included in the Supplementary Material.

**References**

Bisselink, M., B.and Zambrano-Bigiarini, P. Burek, and A. De Roo (2016), Assessing the role of uncertain precipitation estimates on the robustness of hydrological model parameters under highly variable climate conditions, *Journal of Hydrology: Regional Studies*, *(under review)*.

Chen, C.-J., S. U. S. Senarath, I. M. Dima-West, and M. P. Marcella (2016), Evaluation and restructuring of gridded precipitation data over the Greater Mekong Subregion, *International Journal of Climatology*, p. n/a, doi:10.1002/joc.4696.

Coron, L., V. Andréassian, C. Perrin, M. Bourqui, and F. Hendrickx (2013), On the lack of robustness of hydrologic models regarding water balance simulation: a diagnostic approach on 20 mountainous catchments using three models of increasing complexity, *Hydrology and Earth System Sciences Discussions*, *10*(9), 11,337–11,383, doi:10.5194/hessd-10-11337-2013.

CPC-NCEP-NWS-NOAA-USDC (2011), NOAA CPC Morphing Technique (CMORPH) Global Precipitation Analyses, *Tech. rep.*, Boulder CO, doi:10.5065/D6CZ356W, [Last Accessed: 25.Jan.2016].

Crochemore, L., C. Perrin, V. Andréassian, U. Ehret, S. P. Seibert, S. Grimaldi, H. Gupta, and J.-E. Paturel (2015), Comparing expert judgement and numerical criteria for hydrograph evaluation, *Hydrological Sciences Journal*, *60*(3), 402–423, doi:10.1080/02626667.2014.903331.

Dinku, T., P. Ceccato, E. Grover-Kopec, M. Lemma, S. J. Connor, and C. F. Ropelewski (2007), Validation of satellite rainfall products over East Africa's complex topography, *International Journal of Remote Sensing*, *28*(7), 1503–1526, doi:10.1080/01431160600954688.

Dinku, T., S. J. Connor, and P. Ceccato (2010), Comparison of CMORPH and TRMM-3B42 over mountainous regions of Africa and South America, in *Satellite Rainfall Applications for Surface Hydrology*, edited by M. Gebremichael and F. Hossain, pp. 193–204, Springer Netherlands, doi:10.1007/978-90-481-2915-7_11.

Ebert, E. E. (2007), Methods for verifying satellite precipitation estimates, in *Measuring precipitation from space*, pp. 345–356, Springer.

Habib, E., A. Henschke, and R. F. Adler (2009), Evaluation of TMPA satellite-based research and real-time rainfall estimates during six tropical-related heavy rainfall events over Louisiana, USA, *Atmospheric Research*, *94*(3), 373–388, doi:10.1016/j.atmosres.2009.06.015.

Hijmans, R. J. (2016), *raster: Geographic Data Analysis and Modeling*, r package version 2.5-8.

Hossain, F., and G. J. Huffman (2008), Investigating error metrics for satellite rainfall data at hydrologically relevant scales, *Journal of Hydrometeorology*, *9*(3), 563–575, doi:10.1175/2007JHM925.1.

Huffman, G. J., R. F. Adler, D. T. Bolvin, G. Gu, E. J. Nelkin, K. P. Bowman, Y. Hong, E. F. Stocker, and D. B. Wolff (2007), The TRMM multisatellite precipitation analysis (TMPA): Quasi-global, multiyear, combined-sensor precipitation estimates at fine scales, *Journal of Hydrometeorology*, *8*(1), 38, doi:10.1175/JHM560.1.

Legates, D. R., and T. L. DeLiberty (1993), Precipitation measurement biases in the United States, *Journal of the American Water Resources Association*, *29*(5), 855–861, doi:10.1111/j.1752-1688.1993.tb03245.x.

Maggioni, V., P. C. Meyers, and M. D. Robinson (2016), A review of merged high-resolution satellite precipitation product accuracy during the Tropical Rainfall Measuring Mission (TRMM) era, *Journal of Hydrometeorology*, *17*(4), 1101–1117, doi:10.1175/JHM-D-15-0190.1.

Novella, N., and W. Thiaw (2010), Validation of satellite-derived rainfall products over the Sahel, *Wyle Information Systems/CPC/NOAA*, pp. 1–9.

Pechlivanidis, I. G., B. Jackson, H. McMillan, and H. Gupta (2014), Use of an entropy-based metric in multiobjective calibration to improve model performance, *Water Resources Research*, *50*(10), 8066–8083, doi:10.1002/2013WR014537.

Scheel, M. L. M., M. Rohrer, C. Huggel, D. Santos Villar, E. Silvestre, and G. J. Huffman (2011), Evaluation of TRMM multi-satellite precipitation analysis (TMPA) performance in the Central Andes region and its dependency on spatial and temporal resolution, *Hydrology and Earth System Sciences*, *15*(8), 2649–2663, doi:10.5194/hess-15-2649-2011.

Sevruk, B., M. Ondrás, and B. Chvla (2009), The WMO precipitation measurement intercomparisons, *Atmospheric Research*, *92*(3), 376–380, doi:10.1016/j.atmosres.2009.01.016.

Thiemig, V., R. Rojas, M. Zambrano-Bigiarini, V. Levizzani, and A. De Roo (2012), Validation of satellite-based precipitation products over sparsely gauged African river basins, *Journal of Hydrometeorology*, *13*(6), 1760–1783, doi:10.1175/JHM-D-12-032.1.

Thiemig, V., R. Rojas, M. Zambrano-Bigiarini, and A. De Roo (2013), Hydrological evaluation of satellite-based rainfall estimates over the Volta and Baro-Akobo Basin, *Journal of Hydrology*, *499*, 324–338, doi:10.1016/j.jhydrol.2013.07.012.

---

## Author Comment (AC3) · 29 Nov 2016

Our reply to the comments of the referee was uploaded in the form of a PDF supplement.

Please also note the supplement to this comment:
http://www.hydrol-earth-syst-sci-discuss.net/hess-2016-453/hess-2016-453-AC3-supplement.pdf

---

## Author Response (AR1)

Response to reviews on the manuscript **hess-2016-453**
"*Temporal and spatial evaluation of satellite-based rainfall estimates across the complex topographical and climatic gradients of Chile*" by Zambrano-Bigiarini, Nauditt, Birkel, Verbist and Ribbe 2016

January 18, 2017

We would like first to thank the constructive comments and suggestions raised by V. Maggioni, C. Massari and an anonymous referee (AR2). In the next sections we provide a detailed list with the responses to all the remarks pointed out by the ARs, hoping to clarify all the raised issues. We proceed by listing the reviewers' comments (**in bold text**) and then our corresponding reply.

**V. Maggioni (Referee #1)**

**Comments**

**AR1-C1. Although the literature is saturated with articles that validate satellite precipitation products using some ground-based reference, I always welcome this kind of studies, since the use of these products is still limited by the uncertainty associated with them. Moreover, the authors used a complete array of statistics (both continuous and categorical) and focused on a large study area that has not been previously investigated and that presents challenges in satellite precipitation estimation due to its complex terrain. I believe this work addresses issues that are relevant to the HESS readership. The manuscript is well presented and well written.**

AR1-R1
We thank the referee #1 for her comments about our contribution.

**AR1-C2. I would encourage the authors to review an article that I recently published in JHM: Maggioni, V., Meyers, P.C. and Robinson, M.D., 2016. A Review of Merged High-Resolution Satellite Precipitation Product Accuracy during the Tropical Rainfall Measuring Mission (TRMM) Era. Journal of Hydrometeorology, 17(4), pp.1101-1117. In this article, the authors may find studies that have not cited but that may be relevant to compare their findings with previous work. I would also like to invite the authors to join the International Precipitation Working Group (IPWG), which could largely benefit from their insights on the validation of SRE over mountainous regions in South America.**

AR1-R2
We thank the referee for pointing out our attention to the article *Maggioni et al.* (2016). We reviewed it and compared our findings with those mentioned on it. Finally, we already wrote to the IPWG requesting to be considered as potential new members of the group and in November 09th we got an e-mail from Ziad S. Haddad confirming our new membership.

**AR1-C3. It is my understanding that the authors used the TMPA 3B42 research version and not the real-time in their analysis. Can this be clarified in the text?**

AR1-R3
Corrected. In Page 7 of the new manuscript we modified lines 27-29. Moreover, in Page 8 lines 3 and 4 we added the text "*In this work we only used the 3B42_daily product, which was accumulated to monthly, seasonal or annual values depending on the analysis*".

**AR1-C4. I would add the information that all SREs were rescaled to a common 25 km grid in the abstract**

AR1-R4
Done (line 10 in the abstract of the new manuscript).

**AR1-C5. Table 1 shows that the spatial resolution of CMORPH is 0.25deg, but this product is also available on an 8km grid, as stated in the text**

AR1-R5
CMORPH produces global precipitation analyses at very high spatial (8 km) and temporal (30 min) resolution starting from December 2002 until near-present. However, the data used in this work were 3-hr observations at a spatial resolution of 0.25° in NetCDF format. The 3-hr observations were accumulated to daily, monthly, seasonal and annual totals depending on the analysis. There were two reasons for not using the very high-resolution product (8 km, 30 min): i) the webpage used to implement automatic downloading of data (*CPC-NCEP-NWS-NOAA-USDC*, 2011) only offered the possibility to define a spatial subset (i.e. the Chilean extent) for the 0.25x0.25° data, which saved hard disk space for storage,

and ii) the final comparison among SREs was done using a 0.25x0.25° grid, so we preferred to use the data directly from the provider instead of downloading them at 0.07x0.07° and then aggregating to a 0.25x0.25° grid for carrying out the comparisons. The authors are aware of the existence of the webpage `ftp://ftp.cpc.ncep.noaa.gov/precip/global_CMORPH/30min_8km` with data at the very high spatial and temporal resolution. However, it was not possible for us to obtain data files for any period before 2015. We added a clarifying text in page 6, lines 25 and 26, and also added the 0.07° resolution in Table 1 of the new manuscript.

**AR1-C6. I appreciate the effort of considering several statistical metrics to evaluate the SREs. However, can the author discuss whether all these metrics are necessary to assess the performance of these products? Hossain and Huffman (2008) came up with a list of error metrics for evaluating SREs for hydrological applications. How do the statics introduced by the authors compare to that list?**

AR1-R6
Done. We thank the referee for pointing our attention to the article of *Hossain and Huffman* (2008). First, we agree with *Hossain and Huffman* 2008, hereafter HH08, in that *"there is probably not a unique way of representing error completely"*. Similar to them, our work also intends to increase the commonly used metrics (e.g., correlation, bias, RMSE, etc) in order to provide a better assessment of satellite-based precipitation products over the Chilean territory at hydrologically relevant temporal and spatial scales.

HH08 defined the following error metrics:

1. Probability of rain detection ($POD_{rain}$), as a function of rainfall magnitude

2. Probability of no-rain detection ($POD_{norain}$)

3. First- and second-order moments of the probability distribution during false alarms

4. Correlation lengths for the detection of rain ($CL_{rain}$)

5. Correlation lengths for the detection of no rain ($CL_{norain}$)

6. Conditional systematic retrieval error or mean-field bias (when reference $rain > 0$)

7. Conditional random retrieval error or error variance

8. Correlation length for the retrieval error ($CL_{ret}$) (conditional, when $rain > 0$)

9. Lag one autocorrelation of the mean field bias.

Their work was focused on the use of the previous error metrics for simulating stochastic realizations of satellite rainfall with realistic space-time covariance structures. On the other hand, our work was focused on using error metrics that easily convey quantitative information about the overall skill of different SREs in representing the "ground-truth" (KGE and its three components) as well as the skill of each SRE in representing different rainfall intensities ($PC$, $POD$, $fBias$, $FAR$, $ETS$).

Notwithstanding both HH08 and our work computed the probability of detection for different rainfall intensities ($POD_{rain}$), there is a key difference between them:

$POD_{rain} = Prob\{R_{SAT} > 0 | R_{REF} > t_0\}$ , HH08
$POD_{rain} = Prob\{R_{SAT} \in [t_0, t_1] | R_{REF} \in [t_0, t_1]\}$, our work

so, we are not only interested in knowing if a SRE identified that "it was raining" when the reference observation is larger than a user-defined threshold $t_0$, but we are interested in knowing if both the reference rainfall value and the satellite estimate are comparable in its magnitude, i.e., both of them were in the same user-defined interval $[t_0, t_1]$. To summarise the previous point, we agree with HH08 that $POD_{rain}$ and $POD_{norain}$ are important measures to quantify the ability of a SRE to detect the magnitude of the rainfall rate, however we propose to use the same interval of rainfall rates for the reference and the SRE values in order to classify a "Hit". (See Page 15 lines 1-4 in te new manuscript).

Finally, given the increasing use of the Kling-Gupta efficiency ($KGE$) by the hydrological community since 2009 (e.g. *Thiemig et al.*, 2013; *Coron et al.*, 2014; *Pechlivanidis et al.*, 2014; *Crochemore et al.*, 2015; *Bisselink et al.*, 2016) we think that using the KGE value and its three components (correlation, bias and variability) evaluated at daily, monthly, seasonal and annual time scales is easier to interpret and use by hydrologists than measures 3-9 analysed by HH08. (See Page 17, lines 21-26 in the new manuscript).

As a final summary, we recommend the use of $KGE$ and its three components $(r, \beta, \gamma)$ as continuous indices of model performance at different temporal scales, and the use of $POD$ and $fBias$ as categorical indices of model performance for different rainfall intensities (after evaluating $PC$, $POD$, $fBias$, $FAR$ and $ETS$). (See Page 19, lines 7-14 and 19-20 in the new manuscript).

**Anonymous Reviewer #2**

The Authors provided a comprehensive analysis of 7 satellite-based rainfall products over Chile. The analysis was carried on by considering several continuous and categorical scores, taking into account different time scales (daily, monthly, annual and seasonal). Moreover, the assessment was carried on by considering different climatic zones and altitude ranges. The rainfall products were compared with observed rainfall obtained from 366 station over the Chilean territory. Six out seven of the analyzed products used gauge data to calibrate and correct the rainfall estimates. The paper provided useful insights on the quality of satellite-based rainfall estimates over the complex study area. The paper is well written and clear, but I have some comments that I think should be addressed before publication.

**Comments**

**AR2-C1. The analysis was carried on over a 0.25° grid, changing the spatial resolution of some satellite products. Could the Authors explain the regridding procedure? Did they average all the pixels within a 0.25° grid box? I think that some details should be added in the manuscript.**

AR2-R1
The upscaling procedure applied in this work consisted in transferring values from the high resolution raster cells to each one of the 0.25° grid cells, by using bi-linear interpolation as implemented in the `resample` function of the `raster` R package (*Hijmans*, 2016). This upscaling was carried out in order to enable a consistent point-to-pixel comparison between SREs and ground-based data, which is widely used in comparative studies (e.g. *Chen et al.*, 2016; *Thiemig et al.*, 2012; *Novella and Thiaw*, 2010; *Dinku et al.*, 2007). The aforementioned details about the regridding procedure were added in page 10, lines 9-11 of the new manuscript.

**AR2-C2. Do the Authors think that the regridding procedure have any impact in the satellite products performance?**

AR2-R2
In our opinion, to ensure that the regridding procedure has no effect on the final results of the comparison a specific study would be required, by testing different regridding procedures (e.g., bilinear, nearest neighbor, inverse distance, spline, binning, spectral, triangulation) on different time scales (3-hr, daily, monthly, seasonal, annual). However, based on the discussion on `https://climatedataguide.ucar.edu/climate-data-tools-and-analysis/regridding-overview`, we are confident about discarding effects of the regridding procedure on our results, because it was applied only to the daily values of PERSIANN-CCS-adj and CHRIPSv2 products, which are assumed to be smoothly varying variables within each 0.25° grid cell, and therefore the bi-linear interpolation would be an adequate interpolation technique. A short mention to the possible impact of the regridding technique on our results was added in page 10, lines 12-14 of the new manuscript.

**AR2-C3. Why the Authors did not consider real-time products?**

AR2-R3
Thanks for the comment. The following text was added in Page 6 lines 7-10:

*As the most suitable SRE product found in our research will be used for hydrological modelling in selected basins, we focused only on SREs with long data records and good quality data over the Chilean territory, without giving any priority to low-latency (near)real-time products. In addition, comparability of (near)real-time products and station-based calibrated research products is low (Habib et al., 2009; Huffman et al., 2007).*

In addition, the following text was added in Page 7 lines 17-19:

*It should be noted that PERSIANN-CCS-adj is the only (near)real-time product used in this work, and it was considered just because it was especially developed for Chile.*

A comparison of (near)real-time only products is very likely to be subject of future research of our team.

**AR2-C4. Figure 14 description is completely missing from the text.**

AR2-R4
Corrected. In the new version of the manuscript, Figure 14 was renamed to Figure 11 (due to the removal of some figures, see our reply AR3-fC1), and the following paragraph was added in Page 13 lines 13-15:

"*Figure 11 compares monthly and annual time series of precipitation, as estimated by the seven SRE products used in this work against the observed values register at the corresponding raingauge at selected sites for the Far North, Near North, Centre, South and Far South climate macrozones.*

In addition, the new Figure 11 was was also mentioned in Page 15 line 31 and Page 16 line 5.

**AR2-C5. Do the Authors have an idea for the huge overestimation in the Far North region?**

AR2-R5
As the referee mentioned, SREs showed a large overestimation of precipitation values in the hyper-arid Far North (17.5-26.0°S). The following text was added in Page 15 line 31 to Page 16 line 3:

*This can be attributed to the very low precipitation values in that area (some raingauges have annual values lower than 1 mm/year), where one single event not correctly identified can be responsible of 100% of over or underestimation. This overestimation has been observed as well in the scope of other studies (Dinku et al., 2010; Maggioni et al., 2016).*

**Minor comments**

**AR2-mC1. P1, L16: I think that autumn should be changed in spring.**

AR2-mR1.
Corrected.

**AR2-mC2. P5, L32: Please add the GPCC reference.**

AR2-mR2
Added (P5, L32 of the new manuscript).

**AR2-mC3. P6, L15: Please add some reference for the CMORPH European study.**

AR2-mR3
Added (P6, L20 of the new manuscript).

**AR2-mC4. P7, L20: It should be 3B43-V7.**

AR2-mR4
See our reply AR1-R3.

**AR2-mC5. P12, L1-2: I found the same sentence three times. Please check the text.**

AR2-mR5
Corrected. The sentence "*Daily time series in all the 781 stations analysed in this work can be found in the supplementary material*" was mistakenly used four times in the text. Only the sentence in P5 lines 29-30 of the new manuscript was kept.

**AR2-mC6. Section 5 and 6: Please use the same name for the satellite products throughout the manuscript.**

AR2-mR6
Done. The names used for the seven selected products were: CMORPH, CHIRPSv2, 3B42v7, MSWEPv1.1, PGFv3, PERSIANN-CDR, PERSIANN-CCS-adj.

**AR2-mC7. P41, P42: Please check the captions: the last two rainfall classes should be $[10, 40)$ and $> 40$**

AR2-mR7
Corrected.

**AR2-mC8. Table 1: The spatial resolution of CMORPH should be 0.08°**

AR2-mR8
Se our reply AR1-R5.

**C. Massari (Referee #3)**

This study carries out a validation study of seven state-of-the-art SRE products (TMPA 3B42v7, CHIRPSv2, CMORPH, PERSIANN-CDR, PERSIAN-CCS-adj, MSWEPv1.1 and PGFv3) over the complex topography and diverse climatic gradients of Chile. Different temporal scales (daily, monthly, seasonal, annual) are used in a point to-pixel comparison between precipitation time series measured at 366 stations (from sea level to 4600 m a.s.l. in the Andean Plateau) and the corresponding grid cell of each SRE. Different continuous and categorical scores are used to assess the ability of each SRE to correctly estimate precipitation amounts and intensities. Despite literature is plenty of studies about the assessment of the performance of SRE, additional efforts in this direction is very welcome since:

- new products and new releases are produced every year which are not still validated (I would like to thank the authors to use also new datasets like MSWEP).

- it is not clear what are the best performance scores to be used and which is the effect of the benchmark on the final results

- Only by validation studies we can understand the future directions in the improvement of sensors and retrieval algorithms

For the above reasons I think the paper is of interest of the HESS reader. On the whole, I found the paper well written and organized. It contains a lot of material which however has to be better synthetized in some key figures. For this, I suggest the authors to make an effort for presenting less figures or finding alternative plots or tables able to synthetize the results. I also have some major concerns (but nothing that impedes the paper publications) that the authors should address before the paper can be considered publishable on HESS.

**Comments**

**AR3-C1. The first is the discussion of the accuracy and the density of the benchmark dataset. After divided the area in different zones (latitude and elevation) the authors come up with a different number or rain gauges for each zone for each elevation. Since performance scores can be affected from the network density I suggest to report also the number of gauges used for each latitude and altitude zone and discuss more potential effects on the results.**

AR3-R1
We appreciate the comment of the referee. The number of gauges used for each latitude and altitude zone was reported in Figures 5 and 6 in the new version of the manuscript, respectively. The following text was added in Page 16 lines 6-11 as a discussion about the potential impacts on the results:

*It is worth mentioning that the number of stations in each macroclimate zone (Figure 5) and elevation range (Figure 6) is not the same, which hampers an unbiased comparison among different climate and elevation zones. A higher number of stations is located below 1000 m a.s.l (276 out of 366) and in Central and Near South of Chile (223 out of 366). In particular, Figure 4 shows that stations with high number of days with information and located above 2000 m a.s.l. are concentrated in the Far North, making clear the need of additional monitoring stations in elevations above 2000 m a.s.l from the Near North to the southern extreme of Chile.*

**AR3-C2. Higher elevations in winter may be affected by the problem of snow which can be missed by rain gauges. Can the authors can add a comment on that?**

AR3-R2
We agree with reviewer 3. Uncertainties related to high elevation precipitation are extremely high as there are only very few climate stations above 1000 m a.s.l. As these few stations do not have any snow sensor, they cannot provide information on the snow water equivalent. Three stations with snow pillows were built up during the last three years, and measurements are being collected for future comparisons.

Therefore, we expect to quantify these uncertainties by using SREs in hydrological modelling to close the water balance.

We propose to the referee the addition of the following sentence to the new manuscript (new Section 3.3 in Page 12, lines 5-7):

"*Other uncertainties related to precipitation in high-elevation areas can additionally be attributed to missing snow monitoring equipment (i.e., snow scale or pillow to determine the snow-water equivalent), reducing accuracy of precipitation measurements during the winter season.*"

**AR3-C3. It is not clear the rationale behind the choice of the SRE products. Some of them are gauge adjusted while some others not (e.g. CMORPH) despite a gauge corrected version of this product exists. I expect some comments on that.**

AR3-R3
See AR2-R3. A short comment on the rationale behind the selection of the SRE products was added at the beginning of the Section 2.2.2 of the new manuscript (Page 6 lines 7-12).

**AR3-C4. In the outlook section (final section) it reads *Despite continuous improvements of most SRE products, a site-specific calibration is still recommended before any use in hydrological studies. This was evidenced by the PGFv3 dataset that showed a better performance compared to the other products due to the statistical merging of local precipitation observations. This highlights the need for adequate station observation networks to complement the SRE products, due to the dependency of SRE product quality on this ground-truthing.* Here, it is taken for granted that the present rain gauge network provides the best performance in hydrological modelling, and since PGFv3 compares well with it then it will provide the best hydrological simulation. However, there is no evidence (by a hydrological simulation) that this is true and even if can accept that this can be assumed reasonable where the rain gauge density is high it cannot be generalized everywhere, especially because PGF uses the same data of the rainfall network. I think this can make believe the reader that gauge data are the truth while they represent just a good (not in all cases) estimate of the truth rainfall. For instance, if the gauge density is too low the spatial representativeness of the rainfall might be seriously affected. I suggest to discuss more this statement or provide a proof able to demonstrate it.**

AR3-R4
We fully agree with the referee comment. Up to date there is no way to provide fully reliable estimates of areal catchment precipitation, especially with high spatial and temporal resolution. Spatial variability of rainfall is very high and cannot be exactly measured even by a very dense network of gauging instruments.

We propose to add a short Section 3.3 entitled "Uncertainties in the verification results" (Page 12, lines 1-13).

**Additional comments**

**AR3-aC1. Pag. 2. Lines 24-29. MINOR. I think it is not relevant how other people called satellite precipitation estimates.**

AR3-aR1
We partially agree with the referee. The individual name that each research group uses to call satellite precipitation estimates does not make any difference at the moment of evaluating their performance. However, due to the increasing interest in on SRE products, we prefer to provide a short summary of possible names found in literature as a quick guide to those beginners searching for validations or applications of different products.

**AR3-aC2. Pag 4. Lines 10. MODERATE. *The present study attempts to exhaustively evaluate -for the fist time- the suitability of seven state-of-the-art SRE products for hydrological application in this data-scarce and complex mountainous region.* There is no hydrological validation of the SRE in the paper. Please consider revising.**

AR3-aR2
Corrected. We propose to change the previous sentence by:

*"The present study attempts to exhaustively compare -for the fist time- seven state-of-the-art SRE products against ground-based measurements in this data-scarce and complex mountainous region, in order to provide guidelines about suitable SRE products for future hydrological applications"* (page 4, lines 12-14).

**AR3-aC3. Pag. 13 MODERATE. Figure 14 is not cited neither discussed in the text.**

AR3-aR3
Corrected. See AR2-R4.

**AR3-aC4. Pag. 14 lines 28 MODERATE. Annual performance worse than daily ones. There could be a dependence on the number of points (only 8, 2003-2010 for the yearly analysis) and thus a scarce representativeness on the performance of the products? Or it is an effect of the systematic bias a shorter time scale which then is amplified at annual scale. Can you add a comment on that?**

AR3-aR4
We thank the comment and propose the referee to change the current text:

*"which very likely is due to the effect of monthly to annual temporal aggregation, where small differences between monthly SRE values and observations during a large number of months are then summed up when aggregated into a smaller number of annual values"*

by:

*"which was due to the amplification of small systematic biases at the daily time scale when aggregated into the annual one, passing from 2922 days to only 8 annual values"* (Page 16, lines 17-18).

**AR3-aC5. Supplementary material is not used in the text and some performance scores are calculated but never cited in the text.**

AR3-aR5
Corrected. Supplementary material was briefly mentioned in selected sentences of the text (P5 line 30, P12 lines 24-25, P13 line 1, P14 lines 31-32, P15 line 31, P16 lines 3-4) and the $PC$, $ETS$, $fBias$ performance indices (previously not explicitly mentioned in the manuscript) are now analysed in Section 4.3, along with $KGE$, $r$, $\alpha$, $\gamma$, $POD$, $FAR$.

**Comments on the figures**

**AR3-fC1. The number of figures in the paper and in the supplementary material is enormous. I suggest to merge figures 4, 5, 6 and -7 in one figure and better highlight the colour of the dots since they are difficult to read (you can use a different colorbar).**

AR3-fR1
Figures 5, 6 and 7 (in the original manuscript) were moved to the supplementary material, and only the current Figure 4 was, because it summarises the previously mentioned three figures. In addition, previous figures 15 ($POD$) and 16 ($FAR$) were merged into the new Figure 12 (see also our reply AR3-fR3).

Regarding the colorbar, the one used for the figures was selected following the *modified spectral scheme* suggested in the EOS article *The End of the Rainbow? Color Schemes for Improved Data Graphics* ( https://eos.org/features/the-end-of-the-rainbow-color-schemes-for-improved-data-graphics). The authors tried to find a better combination of colours for the new manuscript, but we did not find a better color scheme that would be equally good for normal and color-deficient people. Therefore, we kept the original color scheme but checked that each one of the PDF files included in the supplementary material allowed to make use of different levels of zoom, in order to clearly appreciate differences among point values.

**AR3-fC2. Box plots are interesting but you can also consider to organize figures 10 to 13 in a Table. It will show the same concepts and will allow a better comparison between the different components of the KGE.**

AR3-fR2
In the early stages of this manuscript we started using tables but then we moved into boxplots, mainly because of the very large amount of numbers produced in each table. In particular, each one of the boxplots presented in figures 10 to 13 (in the original manuscript) summarises the empirical distribution

of 366 values for each SRE and time scale, so a table trying to depict a similar amount of information would contain 17934 rows (7 SRE x 366 stations x 7 temporal scales). Therefore, we prefer to keep the boxplots because they visually summarise the information that in a table would require much more space.

**AR3-fC3. POD and fBIAS can be shown in a figure where in the x-axis you have the rainfall classes and on the y-axis you have POD and FBIAS. Then different curves will point to different products.**

AR3-fR3
We thank the comment. Figures 15 ($POD$) and 16 ($FAR$) in the original manuscript were merged into the new Figure 12, which depicts the median values of all the categorical indices of performance ($PC$, $POD$, $FAR$, $ETS$, $fBias$) for different SREs and the five classes of rainfall intensity defined in Table 2.

**References**

Bisselink, B., M. Zambrano-Bigiarini, P. Burek, and A. de Roo (2016), Assessing the role of uncertain precipitation estimates on the robustness of hydrological model parameters under highly variable climate conditions, *Journal of Hydrology: Regional Studies*, *8*, 112–129, doi:10.1016/j.ejrh.2016.09.003.

Chen, C.-J., S. U. S. Senarath, I. M. Dima-West, and M. P. Marcella (2016), Evaluation and restructuring of gridded precipitation data over the Greater Mekong Subregion, *International Journal of Climatology*, p. n/a, doi:10.1002/joc.4696.

Coron, L., V. Andréassian, C. Perrin, M. Bourqui, and F. Hendrickx (2014), On the lack of robustness of hydrologic models regarding water balance simulation: a diagnostic approach applied to three models of increasing complexity on 20 mountainous catchments, *Hydrology and Earth System Sciences*, *18*(2), 727–746, doi:10.5194/hess-18-727-2014.

CPC-NCEP-NWS-NOAA-USDC (2011), NOAA CPC Morphing Technique (CMORPH) Global Precipitation Analyses, *Tech. rep.*, Boulder CO, doi:10.5065/D6CZ356W, [Last Accessed: 25.Jan.2016].

Crochemore, L., C. Perrin, V. Andréassian, U. Ehret, S. P. Seibert, S. Grimaldi, H. Gupta, and J.-E. Paturel (2015), Comparing expert judgement and numerical criteria for hydrograph evaluation, *Hydrological Sciences Journal*, *60*(3), 402–423, doi:10.1080/02626667.2014.903331.

Dinku, T., P. Ceccato, E. Grover-Kopec, M. Lemma, S. J. Connor, and C. F. Ropelewski (2007), Validation of satellite rainfall products over East Africa's complex topography, *International Journal of Remote Sensing*, *28*(7), 1503–1526, doi:10.1080/01431160600954688.

Dinku, T., S. J. Connor, and P. Ceccato (2010), Comparison of CMORPH and TRMM-3B42 over mountainous regions of Africa and South America, in *Satellite Rainfall Applications for Surface Hydrology*, edited by M. Gebremichael and F. Hossain, pp. 193–204, Springer Netherlands, doi:10.1007/978-90-481-2915-7_11.

Habib, E., A. Henschke, and R. F. Adler (2009), Evaluation of TMPA satellite-based research and real-time rainfall estimates during six tropical-related heavy rainfall events over Louisiana, USA, *Atmospheric Research*, *94*(3), 373–388, doi:10.1016/j.atmosres.2009.06.015.

Hijmans, R. J. (2016), *raster: Geographic Data Analysis and Modeling*, r package version 2.5-8.

Hossain, F., and G. J. Huffman (2008), Investigating error metrics for satellite rainfall data at hydrologically relevant scales, *Journal of Hydrometeorology*, *9*(3), 563–575, doi:10.1175/2007JHM925.1.

Huffman, G. J., R. F. Adler, D. T. Bolvin, G. Gu, E. J. Nelkin, K. P. Bowman, Y. Hong, E. F. Stocker, and D. B. Wolff (2007), The TRMM multisatellite precipitation analysis (TMPA): Quasi-global, multiyear, combined-sensor precipitation estimates at fine scales, *Journal of Hydrometeorology*, *8*(1), 38, doi:10.1175/JHM560.1.

Maggioni, V., P. C. Meyers, and M. D. Robinson (2016), A review of merged high-resolution satellite precipitation product accuracy during the Tropical Rainfall Measuring Mission (TRMM) era, *Journal of Hydrometeorology*, *17*(4), 1101–1117, doi:10.1175/JHM-D-15-0190.1.

Novella, N., and W. Thiaw (2010), Validation of satellite-derived rainfall products over the Sahel, *Wyle Information Systems/CPC/NOAA*, pp. 1–9.

Pechlivanidis, I. G., B. Jackson, H. McMillan, and H. Gupta (2014), Use of an entropy-based metric in multiobjective calibration to improve model performance, *Water Resources Research*, *50*(10), 8066–8083, doi:10.1002/2013WR014537.

Thiemig, V., R. Rojas, M. Zambrano-Bigiarini, V. Levizzani, and A. De Roo (2012), Validation of satellite-based precipitation products over sparsely gauged African river basins, *Journal of Hydrometeorology*, *13*(6), 1760–1783, doi:10.1175/JHM-D-12-032.1.

Thiemig, V., R. Rojas, M. Zambrano-Bigiarini, and A. De Roo (2013), Hydrological evaluation of satellite-based rainfall estimates over the Volta and Baro-Akobo Basin, *Journal of Hydrology*, *499*, 324–338, doi:10.1016/j.jhydrol.2013.07.012.